# SEMORE: SEgmentation and MORphological fingErprinting by machine learning automates super-resolution data analysis

Steen W. B. Bender [1,2,3], Marcus W. Dreisler [1,2,3], Min Zhang [1,2,3], Jacob Kæstel-Hansen [1,2,3] ✉ & Nikos S. Hatzakis [1,2,3,4] ✉

The morphology of protein assemblies impacts their behaviour and contributes to beneficial and aberrant cellular responses. While single-molecule localization microscopy provides the required spatial resolution to investigate these assemblies, the lack of universal robust analytical tools to extract and quantify underlying structures limits this powerful technique. Here we present SEMORE, a semi-automatic machine learning framework for universal, system- and input-dependent, analysis of super-resolution data. SEMORE implements a multi-layered density-based clustering module to dissect biological assemblies and a morphology fingerprinting module for quantification by multiple geometric and kinetics-based descriptors. We demonstrate SEMORE on simulations and diverse raw super-resolution data: time-resolved insulin aggregates, and published data of dSTORM imaging of nuclear pore complexes, fibroblast growth receptor 1, sptPALM of Syntaxin 1a and dynamic live-cell PALM of ryanodine receptors. SEMORE extracts and quantifies all protein assemblies, their temporal morphology evolution and provides quantitative insights, e.g. classification of heterogeneous insulin aggregation pathways and NPC geometry in minutes. SEMORE is a general analysis platform for super-resolution data, and being a time-aware framework can also support the rise of 4D super-resolution data.

Biomolecules form diverse assemblies with high spatial ordering such as clusters, biomolecular condensates or aggregates, both during regular and aberrant cellular function. The morphology of these diverse assemblies, i.e., geometry, topology, size, shape and internal structure can have a significant impact on their properties and often inhibit their functions[1]. Examples include heterogeneous assemblies for metabolons[2] and signalosomes[3,4], genome[5,6], phase-separated or membrane-less organelles[1,7] and nuclear pore complexes[8,9] but also protein aggregates which underlie many neurological disorders[10–13].

Various approaches including ensemble[14–18] and single molecule studies[19–23] have been employed to study the morphology of

biomolecular assemblies. The implementation of Single-Molecule Localization Microscopy (SMLM) has revolutionized cell biology by capturing biological assemblies at nanoscale spatial resolution in biological samples. Stochastic Optical Reconstruction Microscopy (STORM)[24], PhotoActivated Localization Microscopy (PALM)[25,26], Points Accumulations for Imaging in nanoscale Topography (PAINT[27] and DNA-PAINT[28]) as well as REal-time kinetics via binding and Photobleaching Localization Microscopy (REPLOM)[19] are SMLM techniques that surpass the diffraction limit by individual molecule localization through fluorescent probes. For both STORM and PALM, these localisations are achieved through photo switchable blinking, for

[1]Department of Chemistry, University of Copenhagen, Copenhagen, Denmark. [2]Center for 4D cellular dynamics, University of Copenhagen, Copenhagen, Denmark. [3]Novo Nordisk Center for Optimised Oligo Escape and Control of Disease, University of Copenhagen, Copenhagen, Denmark. [4]Novo Nordisk Center for Protein Research, University of Copenhagen, Copenhagen, Denmark. ✉e-mail: jkh@chem.ku.dk; hatzakis@chem.ku.dk

PAINT by the reversible association with the target. REPLOM utilizes the kinetic behaviour of self-assembly systems and photobleaching by surface docking in a photo-unstable environment, unlocking the temporal resolution of self-assembly kinetic pathways. Despite progress in acquiring these information-rich data sets, the analysis and identification of individual protein assemblies in SMLM are often reliant on manual annotations or system-specific approaches which are resource and time-strenuous and lack generalisation[29–31].

The advancement of machine learning-based approaches[32–42] has been instrumental for quantitative image analysis[30,43,44] and has the potential to resolve the bottleneck of extraction of assemblies of interest in super-resolution data. In general, these approaches can be broadly categorised as either supervised or unsupervised, each of which has advantages and limitations[30]. Supervised algorithms are highly accurate when large amounts of annotated data are available albeit annotations require extensive manual labor and expert knowledge and the resulting model is often suitable for one specific data set or task. This imposes some challenges in exploring unmapped biological systems with no a priori knowledge and potentially limits their use as a general tool[41,45,46]. Unsupervised approaches such as OPTICS[47] and DBSCAN[48] can overcome some of these limitations for coordinate-based input data. Their performance however is often limited by a one-size-fits-all approach. This often results in laborious human intervention in model tuning, restricting their adaptation to heterogeneity in localization densities and assembly sizes in varying experimental data[43,49]. While these approaches support multidimensional data input, temporal information is not incorporated into distance coordinate systems, rendering clustering algorithms ineffective for handling a temporal axis. The above challenges limit the robust, generalised extraction of protein assemblies' geometry and kinetics across systems in SMLM.

Here we present SEMORE, an unsupervised machine learning pipeline that allows the rapid agnostic and precise transformation from raw spatiotemporal localization SMLM data into individualized protein assemblies by mapping their diverse morphologies by an extensive set of descriptive features. We show that SEMORE provides unbiased unsupervised clustering, and morphological cluster variation in time, without a priori knowledge and for diverse simulated and experimental data sets: heterogeneous growth pathways of insulin aggregates, the dimensions of individual nuclear pore complexes, size of individual clusters of fibroblast growth receptors 1, temporal evolution of syntaxin 1a clusters and dynamic clustering of ryanodine receptors (RyR). The implementation of temporal dependence in morphological variations is a promising platform to handle static or dynamic super-resolution data and enables in-depth temporal-dependence analysis and segmentation of complex structures.

## Results

### SEMORE: Segmentation and Morphological Fingerprinting

SEMORE's architecture consists of two main modules: a clustering module and a morphological fingerprinting module. The model can accept as input any set of x, y, coordinates (Fig. 1a) that is routinely produced in super-resolution approaches such as STORM, PALM, PAINT and DNA-PAINT[29,50,51]. Additionally, SEMORE has the capability to incorporate temporal information, allowing for the processing of x, y, and t (time-resolved) localization data, as introduced by recent methods such as REPLOM, that included the temporal dimension in SMLM[19]. Utilizing its two independent modules and requiring minimal

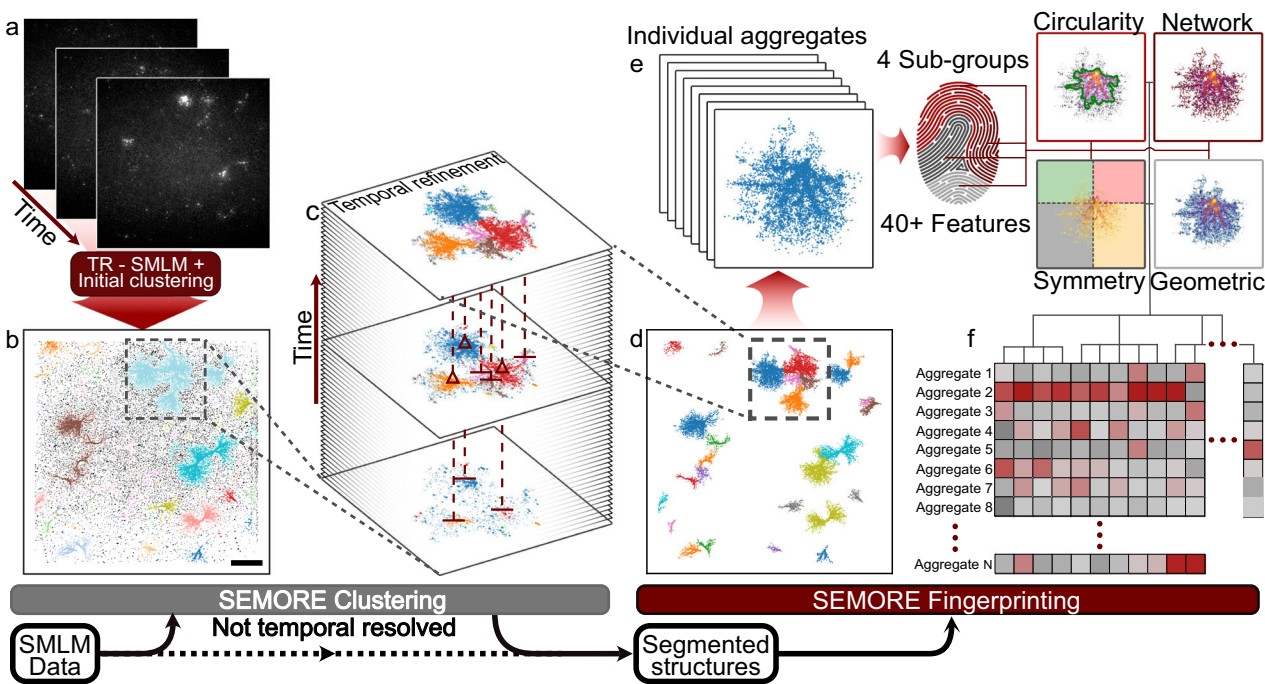

**Fig. 1 | Schematic illustration of SEMORE, an automated pipeline to agnostically cluster and classify temporarily and morphologically distinct protein aggregates. a** SEMORE input is a set of x, y SMLM input (PALM or STORM) or x, y, t time-resolved SMLM (TR-SMLM) (REPLOM input) coordinates of individual localization/aggregation events, here shown for temporally resolved insulin aggregation imaged using the REPLOM[19] approach on a TIRF microscope. **b** The first step of SEMORE clusters data by a density-based clustering method in three dimensions of the spatial coordinates, x and y, and time, t. Colours indicate clusters and scale bar shows 10 μm. **c** The second step is a temporal refinement of the initial clusters to identify and dissect underlying sub-clusters, utilizing a time-directional clustering through the iteration of frames. **d** The final output of the temporal refinement is a set of individual spatially resolved structures that are now separable even if grown close to other aggregations. **e** Each identified cluster is fed to a morphology-fingerprinting module that computes four groups of descriptive features, including circularity of the morphology, graph network within the aggregation, general symmetry, and geometric interior. Combined these feature groups construct the individual self-assembly fingerprint of a total of 40+ features. **f** The calculated morphology fingerprints are stored for each extracted protein assembly allowing for complete quantification and insights into the distribution of heterogeneous morphology or growth pathways. Source data are provided as a Source Data file.

human intervention, SEMORE outputs a list of individualized clusters along with their complete morphological quantification (Fig. 1d).

The clustering module of SEMORE consists of multiple steps that self-parameterize based on the input data and are scale invariant due to the inherent 3D axis standardization (see Methods). This allows SEMORE to operate independently of the imaging and cluster dimensions and achieve high classification accuracies solely based on the number of localizations contained within the protein assembly given the spanning area. This is designed to account for the inherently heterogeneous nature of biological assembly systems, in size, scale, spatial overlap, density, and morphology as well as the variability across experimental configurations that challenges current analytical tools. The pipeline initially inspects high-density areas in a standardized Euclidean 3D space, using a hyperparameter space pre-defined for this region, and provides an appropriate model choice based on a data-driven decision. The chosen density-based scanning model, either HDBSCAN or DBSCAN, extracts high-density regions of biomolecules (clusters or aggregates) from low-density regions (noise) (Fig. 1b). The initial clustering contains an added topological fail-safe to prevent the detection of nonsensical structures (see Methods). If a temporal dimension is available, the high-density regions are treated through our temporal refinement (Fig. 1c). Segmentation in time and temporal refinement is strictly required to dissect spatially overlapping structures within high-density areas. The clustered output is further refined by subjecting all identified assemblies to a smart density filter to eliminate falsely predicted assemblies that do not meet agnostic, data-derived density criteria (Supplementary Fig. 1). The result is a robust clustering model outcompeting current methods and building, to the best of our knowledge, towards the first general-purpose approach for dynamic SMLM (see Methods and Supplementary Fig. 4).

The morphological fingerprinting module of SEMORE is designed to capture inherent structural variations and morphological diversity across different structure types or even within the same structure family. Fingerprinting has been employed as unique identifiers in signal processing[52], genetics[37] and recently to dissect and predict the identity of heterogeneous diffusion[38]. In SEMORE, the fingerprinting module takes individual assemblies and generates a unique fingerprint consisting of 40+ descriptive features. They are based on circularity, symmetry, graph network statistics and geometric densities (see Fig. 1e, Supplementary Table 1, and Methods). Circularity (5 features): Describes different estimators for circular resemblance, resulting in a multivariate feature-set, depicting ratios of circular properties. Symmetry (7 features): extracts the balance of points occurrences and extent around the 2D-axis with origin at the structure middle. Graph networks (25+ features): includes density and edge-based characteristics such as "longest-shortest path" that provides structural insight for simple or complex as well as convex or concave bounding polygons. Geometric (3 features): retrieves the average density aspect and an estimate of the furthest connecting location pairs, offering precise size estimations. The number and identity of features are constructed as diverse as possible so as to agnostically capture a diverse set of protein clustering morphologies and maximize applicability across biological systems without a priori knowledge. Note that some of the features can have overlapping interpretations. This overlap does not affect the overall method accuracy as only features that enhance the classification are taken into account, however, they can collectively contribute to a detailed description of the system at hand. If better features are identified in the future, they can be conveniently implemented into SEMORE further extending its potential. Besides providing statistical insights, the morphology fingerprinting module enables the clustering of distinct self-assembly structures and identifies morphology diversity within the same structure family. The latter is particularly important given our recent study on time-resolved protein aggregation found insulin exhibits diverse aggregation growth mechanisms: anisotropic and isotropic[19,53].

## Accurate extraction of individual assemblies across diverse biologically inspired growth types

To assess SEMORE's performance for structure extraction in noisy environments, we simulated 3 biologically inspired aggregational types: 1) Symmetric isotropic growth; representing high-density spherical structures with a density drop at the edges of the structure (Fig. 2a), inspired by isotropic-spherulite growth in biological and physical systems[54,55] (see Methods). 2) Sterically driven growth; representing more asymmetrical random growth where structures may branch and create morphologically distinct assemblies based on minimizing steric hindrance (second-row Fig. 2a). 3) Fibril growth; depicting the growth of thin and branching fibrils commonly seen in alpha-synuclein aggregation[56] (third-row Fig. 2a). After the structures have been simulated, noise is either uniformly added in 3 dimensions (x, y, t) to the whole field of view or added as heterogeneous noise from randomly placed noise seeds (see Methods) These different structures are simulated 50 times in a 40×40 μm FOV each with 10, 10 and 25 individual aggregates in each simulated movie. Their start and end frames as well as the growth extends of the structures are randomly drawn (see Methods), often resulting in overlapping and heterogeneous structures resembling real data (Fig. 2a).

SEMORE precisely classified all 3 types of morphologies in this stress test with F1 scores ranging from 84–98% even in very noisy conditions. More specifically, isotropic growth classification reached a median accuracy of 83% and an F1 score of $84 \pm 4\%$. The confusion matrices in Fig. 2b displays the agreement between the ground truth and the predicted labels. Most misclassifications stem from the low-density edges of these isotropic assemblies hidden in noise and in comparison, the simpler approach DBSCAN/HDBSCAN achieved $5 \pm 1\%$/ $36 \pm 10\%$ accuracy with a $6 \pm 2\%$/ $40 \pm 10\%$ F1 score. Sterically driven aggregation classification accuracy of $83 \pm 7\%$ and an F1 score of $90 \pm 3\%$. Again, most misclassification stems from the edges of these assemblies and the competing DBSCAN/HDBSCAN achieves $40 \pm 10\%$/ $22 \pm 6\%$ mean accuracy with an F1 score of $43 \pm 11\%$/ $26 \pm 7\%$. Fibril classification reached a $94 \pm 2\%$ accuracy and an F1 score as high as $98 \pm 1\%$ even for overlapping fibrils (see Fig. 2). The overlap of fibrils demanded a more conservative search range for the initial clustering but showcases utility and strength of temporal refinement as DBSCAN/HDBSCAN achieved $54 \pm 13\%$/ $30 \pm 7\%$ mean accuracy and $60 \pm 14\%$/ $40 \pm 8\%$ F1 score. The median classification accuracy of SEMORE remained above 85% for a range of noise density levels and was practically independent of noise being homogeneous or heterogeneous (see Methods, Supplementary Fig. 2 & 3). While at extremely low noise ratios i.e., ten times lower than signal or no noise, the smart density filter can result in the removal of true positives, we recommend using the full SEMORE pipeline for data with noise (Supplementary Fig. 3). Unsupervised clustering often requires hyperparameter tuning however the clustering module of SEMORE provides highly accurate unsupervised classification with minimal human tuning (see Methods) and manages to separate overlapping structures of all simulated aggregation types demonstrating its versatility and its broad applicability. Importantly, the temporal element of SEMORE allows the identification of fibrils growing within spherulites, which is otherwise impossible with current state-of-the-art approaches (see Supplementary Fig. 4).

To further evaluate the performance of SEMORE on segmentation and analysis of dynamic SMLM data we performed a series of stress tests on simulated data. We first evaluated SEMORE's ability to track morphological changes in time using simulated data of 3 types of spatially overlapping protein clustering morphologies with temporal information included. Snapshots of SEMORE's clustering in time provide visual confirmation of SEMORE's ability to track morphological changes in time (see Supplementary Fig. 5 for snapshots of simulated and Supplementary Fig. 6 for experimental data). Further quantification of SEMORE's ability to accurately track and segment spatially

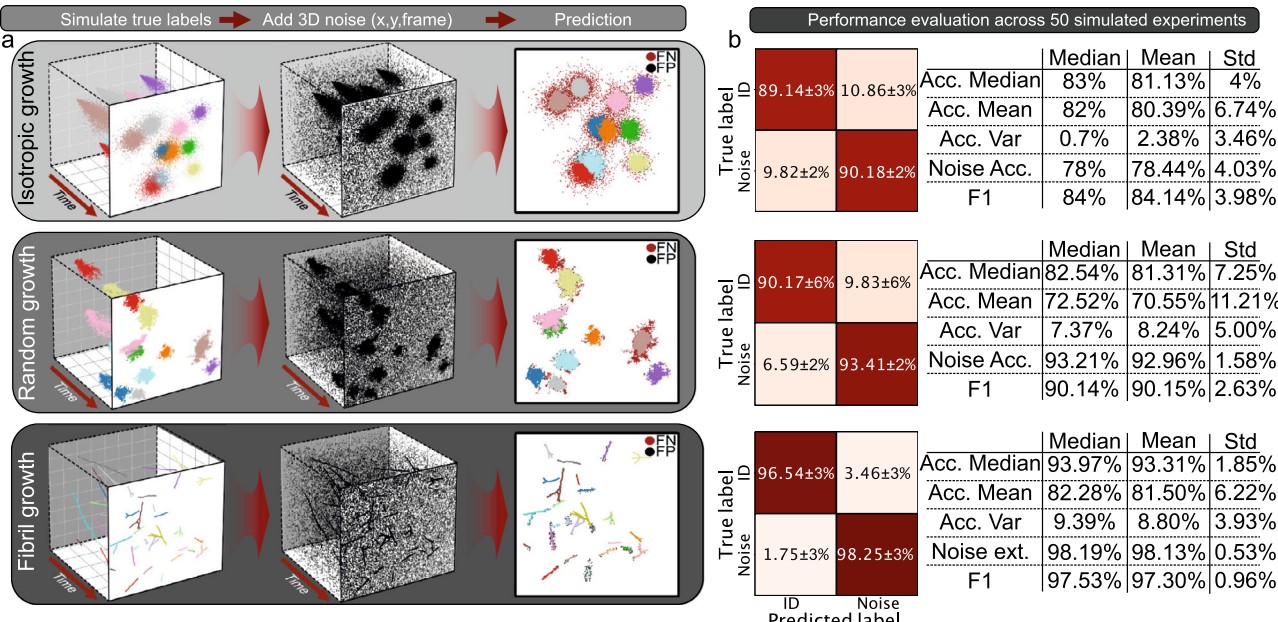

**Fig. 2 | Performance evaluation of SEMORE clustering module on classification of three diverse types of morphologies inspired by biological systems. a** Three classes of time-resolved aggregations were simulated to capture a broad aspect of biological systems (see Methods): isotropic, where aggregates grow radially, where aggregates grow in response to steric hindrance and branching fibrils where aggregates grow linearly followed by branching. The three inserts depict the general pipeline for cluster identification: From left to right Aggregates with diverse final morphologies are produced in a frame-by-frame manner, with the amount and locations of particles randomly drawn based on previous localizations and start and end times randomly drawn. Uniform noise is added in all three dimensions (x, y, time). The model accurately predicts diverse aggregates, showcased by different colours. The black point corresponds to data points predicted as the wrong label, i.e., either noise predicted as an aggregate point or multiple predicted aggregates for the same ground truth label (FP) while the brown points correspond to aggregational locations predicted as noise (FN). **b** Quantification of operational performance by a confusion matrix. Predictions are shown from 50 experiments for each aggregation type, each containing 10 individual aggregations for isotropic and random, and 25 for fibril growth. Errors are standard deviations calculated across accuracies for each individual aggregate. Common classification metrics for the evaluation are shown in the table on the right side of the corresponding confusion matrix. Source data are provided as a Source Data file.

overlapping protein assemblies in time reveals growth onset times across 3 morphological classes are predicted with an average offset of just -13 frames (Supplementary Fig. 6). Subsequently, we test the clustering module on simulated, sparse structures containing as little as 4, 8, 15 and 25 point detections akin to data of protein oligomerization (see Supplementary Figs. 7 and 8). SEMORE analyses the entire field of view at once and extracts structures down to 4 detections while maintaining >90% accuracy at biologically relevant noise-to-signal ratios. The morphological fingerprinting module can further refine the false positive detections from noise with as little as 4 detections, achieve full separation from noise at 8 detections and classify morphological classes in true structures at 15 detections (see Supplementary Figs. 8 and 9).

Lastly, we evaluated its performance on degenerative structures that shrink in time, akin to protein de-polymerisation. SEMORE accurately segments 3 anisotropically degenerative morphologies showcasing it can be used to analyze dynamic shrinkage or depolymerisation of protein clusters (see Supplementary Fig. 10). In essence SEMORE only requires <10 detections to accurately segment and classify heterogeneous structures with dynamic morphologies further demonstrating its operational utility and potential for 4D SMLM.

**Morphological fingerprinting captures defining features separating heterogeneous assemblies**

To demonstrate the utility of the morphological fingerprinting module of SEMORE, each of the extracted aggregates by the clustering module was transformed into a morphological fingerprint. This resulted in unique fingerprints quantifying each of the 2256 individual aggregates and their respective feature ranking, offering mechanistic insights on the morphology features that allow identity dissection (see Fig. 3 with additional simulated examples in Supplementary Figs. 7–9 & 11–15 and the experimental examples in Supplementary Figs. 22 and 23). To obtain a visual and qualitative representation of the morphological fingerprints we utilized a Uniform Manifold Approximation and Projection (UMAP) followed by a Density-Based Spatial Clustering of Applications with Noise (DBSCAN). Dimensionality reduction of the embedded morphological fingerprints displays a well-defined separation of the three simulated growth types into distinct clusters (Fig. 3a, b).

Inspection of the clusters reveals that in addition to grouping extracted aggregates, SEMORE also captures a fourth-class containing noise, (Fig. 3b, c). The noise detections are induced by loosened restrictions in the smart density filtering (see Methods) in the fibril simulations to evaluate performance in challenging experimental settings with imperfect segmentation steps. The resulting UMAP from an unperturbed smart density filtering is depicted in Supplementary Fig. 12. The noise isolation in the embedded space showcases the strength of the morphological fingerprinting module in structure classification and its potential as a post-processing module. Classification performance reaches >99% accuracy, (Fig. 3c, d) showcasing that the noise correction dramatically improves classification accuracy and allows for quantification of classification performance by viewing each identified cluster in the embedded fingerprint space as a prediction.

We evaluated how morphological fingerprinting can dissect the diverse types of otherwise similar morphologies by three approaches. Firstly, utilizing a second UMAP embedding and DBSCAN of the circularity feature subset offered additional investigation of the fibril cluster in the embedded fingerprint space (see Fig. 3b). This resulted

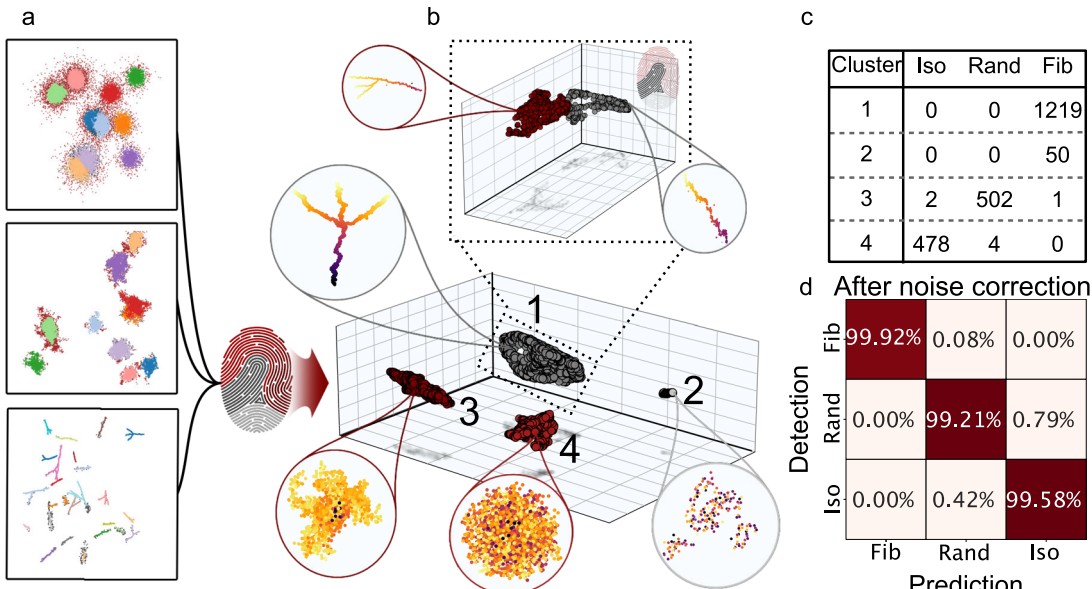

| Cluster | Iso | Rand | Fib |
|---------|-----|------|-----|
| 1 | 0 | 0 | 1219 |
| 2 | 0 | 0 | 50 |
| 3 | 2 | 502 | 1 |
| 4 | 478 | 4 | 0 |

**Fig. 3 | Performance evaluation of Morphology fingerprinting module on three diverse assembly morphologies. a** The three diverse morphological structures of Fig. 2 are subjected to the morphology fingerprinting module. Each colour represents a cluster but brown-red that represents noise detections. **b** The derived features are dimensionality reduced by a 3-component UMAP to visualize the separation of the identified clusters in the latent space and the grouping of the diverse morphologies. The dimensionality-reduced features are clustered using DBSCAN to identify groups of fingerprints. The four identified cluster groups are displayed, corresponding to three different simulated aggregational structures, as well as a cluster containing only pure noise. (Spherical zoom, points coloured by frame value), Further analysis of the group corresponding to fibrils by an additional 3-component UMAP and a new DBSCAN (square dashed line zoom on top), identified two local clusters mainly containing branched and non-branched fibrils respectively (see Supplementary Fig. 13) (spherical zoom, points coloured by frame value). **c** The count of each simulation type is found through a simple investigation of clusters 1 to 4, where cluster 2 only contains data from the fibril simulation and is deemed noise by visual inspection. **d** Confusion matrix of classification accuracy for each cluster after the removal of noise, Cluster 1 predicts fibril (sensitivity 99.92%, F1 99.96%), 3 random (sensitivity 99.21%, F1 99.31%), and 4 isotropic growth (sensitivity 99.58%, F1 99.38%), resulting in an average F1 score at 99.55 ± 0.21%, clearly showing the descriptive information of morphology captured within the fingerprinting. Errors are standard deviations calculated across all aggregates. Source data are provided as a Source Data file.

in two spatially separated clusters corresponding to branching and non-branching fibrils which independently can be achieved by boosted decision tree classification using all fingerprint features directly (see Supplementary Fig. 13). Secondly, by investigating the isotropic and anisotropic clusters which revealed more continuous spaces given their more smooth growth behaviours as compared to branching of fibrils, yet with clear spatial separation of fingerprint features (see Supplementary Fig. 14). Lastly, SEMORE was able to correctly classify the identity of diverse morphologies in high-density regions reaching an F1 score of >98% (See Supplementary Fig. 11). Although, morphological fingerprints represent unsupervised output these results demonstrate their versatility in supervised classification to extend beyond distinguishing between fundamentally different morphology families, i.e., fibrils vs isotropic, to also capture heterogeneity within the same morphology family i.e., branching fibrils vs non-branching fibrils. We find the morphological fingerprinting needs just 8 detections to fully separate true detections from noisy and 15 detections to further classify the morphology class of the true detections (Supplementary Figs. 7 and 8). Such expressive power is required to provide mechanistic insights for most biological assemblies as they often follow one assembly mechanism but still exhibit heterogeneity in their final morphologies and the mapping of which is currently an analytical challenge[57].

A central element of SEMORE is that it inherently performs temporal segmentation thus offering the potential to capture gradual morphological changes in super-resolution data. To evaluate SEMORE's performance we simulated dynamic morphology variation between three major morphologies (fibril-like, isotropic, and sterically-hindered) (Supplementary Figs. 15 and 26). We simulated thirty structures of each morphology class (totalling 90), placed these sequentially in random order whilst ensuring no identical morphology consecutively. Between each of the ninety structures 100 positions are constructed by interpolation (see Methods) resulting in 8900 intermediate structures. UMAP representation in Supplementary Fig. 26 shows SEMORE accurately captures distinct morphology classes and reliably tracks their gradual dynamic morphology change by placing intermediate structures on a gradient between the distinct morphology classes of where the transition is happening. Note, the UMAP is simply a visualization tool to show the structure of the high dimensional data manifold of the 8900 morphological fingerprints, it is not a requirement for usage and may vary for specific cases. As these structures exist in a continuous space SEMORE allows future users to identify dynamic morphological variations and decide on system-specific decision boundaries for each of the structures.

Blinking is a common challenge in SMLM, especially for PALM and dSTORM, and may lead to artificial clustering[29] and misinterpretation of protein assembly morphology. We map the effect of blinking on the morphological fingerprinting module using data with and without simulated blinking (see Methods). SEMORE remains largely unaffected by blinking achieving segmentation accuracies above 90% and reliably differentiates morphological classes from just 15 detections regardless of blinking (see supplementary Fig. 9).

In addition to blinking, protein assemblies can exhibit biological behaviour involving significant morphological changes (e.g., spherical to tubular), or as discussed in the evaluation of the clustering module above, depolymerisation. To this end, we evaluated SEMORE's ability to track the temporal evolution of morphology between diverse morphologies, i.e., from fibril-like to spherical to asymmetric. SEMORE clearly tracks morphologies in time and transitions while outputting the most distinguishing features of each morphology offering important mechanistic insights (supplementary Fig. 15).

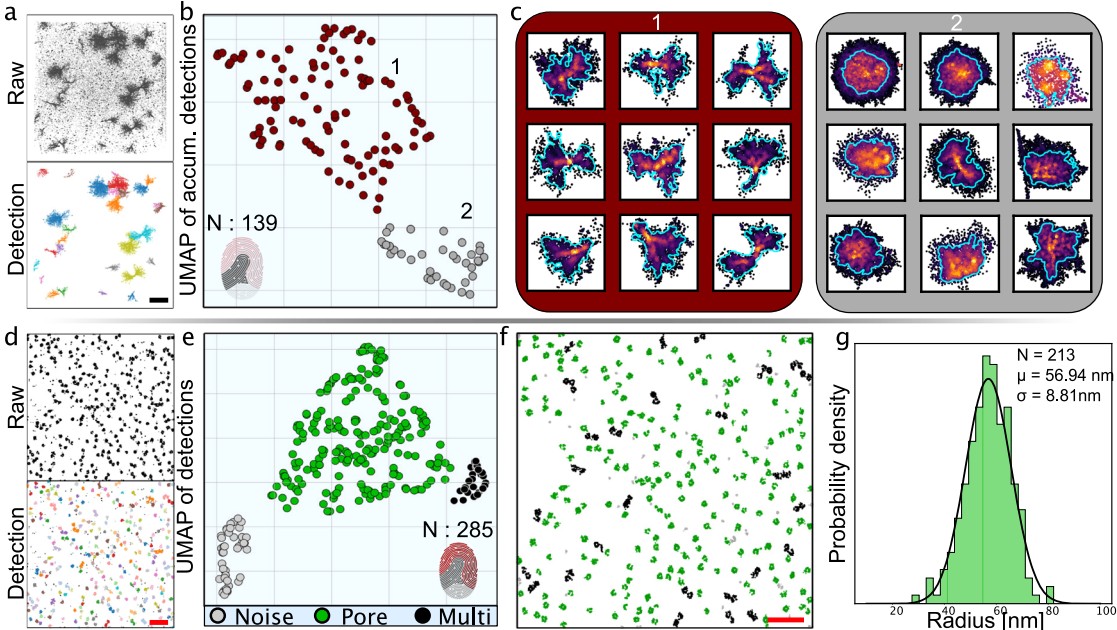

**Fig. 4 | The SEMORE pipeline generalizes across widely diverse experimental systems, time-resolved insulin aggregation and Nuclear Pore complex (NPC) assembly. a** Top: Final frame of accumulated super-resolution localizations from temporally resolved insulin aggregation. Bottom: Identification of each aggregate depicted as a distinct colour and calculation of its corresponding fingerprint by SEMORE. The scale bar shows 10μm. **b** The collective fingerprints are processed through a 2-component UMAP and clustered using DBSCAN, resulting in two clusters: cluster 1 (red) contains low-density elongated anisotropic growth patterns, and cluster 2 (gray) contains isotopically grown high-density spherical-like structures. **c** Nine representative aggregates for each of the anisotropic and isotropic clusters, with points coloured by frame value. **d** Top: Accumulated super localizations of NPC assemblies from ref. 58. Bottom: Identification of each assembly depicted as distinct colour and calculation of its corresponding fingerprint by SEMORE. The scale bar shows 1μm. **e** Processed fingerprints of NPC and 2-component UMAP and clustered using DBSCAN in 3 clusters: Cluster 1 (green) corresponds to individual NPC assemblies, cluster 2 (black) to overlapping NPC assemblies and cluster 3 (gray) to noise. **f** Overlay of the clustered NPC color-coded based on their classification. Scale bar shows 1μm. **g** extracted radius of NPC consistent with earlier reports[63]. Source data are provided as a Source Data file.

## Precise extraction and quantification of experimental super-resolution data

To evaluate how SEMORE performs and generalizes on experimental data across experimental systems we tested its efficiency on two diverse, yet representative, data structures found in SMLM: Time-resolved spatiotemporal localizations of insulin aggregates by REPLOM[19] and typical, static super-resolution data from dSTORM of nuclear pore complexes (NPC)[58] and fibroblast growth receptor 1 (FGFR1)[35] as well as dynamic live-cell PALM of ryanodine receptors (RyRs)[59] and sptPALM of syntaxin 1a (Sx1a)[60].

Studies of aggregating insulin using the super-resolution approach REPLOM, offering real-time recordings of the aggregation growth, revealed insulin aggregates by two distinct pathways: an isotropic pathway, generating radially grown spherical structures and an anisotropic pathway, generating more elongated asymmetric structures consistent with the presence of multiple nucleation sites. When the raw x, y, t coordinates of insulin aggregation events were fed to SEMORE, the clustering module automatically extracted 139 individual insulin aggregate structures (Fig. 4a) despite them displaying large heterogeneity in their morphology. The identified structures are largely spatially overlapping, an element that would challenge their proper separation by manual inspection or current tools. However, the implementation of the temporal element of SEMORE capturing time-dependent morphological evolution allowed their rapid and precise spatial separation, thus mitigating potential morphology misinterpretation (Fig. 4a and Supplementary Fig 5). Snapshots of the successful capture of insulin aggregation during its growth combined with the insulin aggregation initiation time point identification by SEMORE are displayed in Supplementary Fig. 6. The fingerprinting module of SEMORE provided 2 well-defined clusters of insulin aggregates as identified by DBSCAN (eps = 1) (Fig. 4b) (see Methods). Cluster 1 contains sparse elongated morphologies indicative of anisotropic structures while Cluster 2 contains dense isotropically grown structures in agreement with published results[19]. SEMORE required a few minutes to correctly and agnostically identify all 139 clusters as compared to days by manual annotation by experts and did so automatically, without any hyperparameter tuning. This supports that SEMORE offers agnostic insights on heterogeneous growth pathways and morphologies of actual insulin aggregation automatically, with minimal human intervention, accelerating data analysis from weeks to minutes. Feature ranking (see Methods and Supplementary Fig. 16) on the other hand, offers users to inspect the distinguishing features of heterogeneous growth pathways to derive mechanistic insights.

We further demonstrated SEMORE's ability to utilize the temporal dimension in experimental data using three additional published data sets: Firstly, dSTORM data of fibroblast growth receptor 1[35], where SEMORE utilizes the data's reported frames for temporal refinement together with smart density filtering to accurately extract protein clusters, prune false positive detections and provide cluster size estimation (Supplementary Fig. 17). Secondly, super-resolved dynamic data syntaxin 1a spatial clustering by sptPALM[60], where SEMORE accurately captures clusters, their morphological evolution in time (Supplementary Fig. 18). Thirdly, super-resolved live-cell PALM data of ryanodine receptors (RyRs)[59]. SEMORE accurately captures RyR clusters with a granularity infeasible by current methods such as DBSCAN alone (Supplementary Fig. 19). While in experimental data sets it is hard to strictly define ground truth SEMORE outputs qualitatively outputs identical predictions with current states of the art. The fact that SEMORE operates across dynamic experimental SMLM data outputting clusters and their properties in agreement with published analysis, further highlights its potential as a robust universal tool for 4D cellular biology.

Finally, we tested SEMORE's ability to extract assemblies and perform quantification on a completely different system: super-resolution imaging by dSTORM of Nuclear Pore Complexes (NPC)[58]. The data set consists of static images recorded by dSTORM, only requiring the SEMORE's morphological fingerprinting module. Through a simple DBSCAN, 285 segmentations were extracted and directly fed into the SEMORE pipeline. The fingerprinting module outputs per NPC an individual unique morphological fingerprint resulting in three distinct clusters in the UMAP embedded space: Individual NPCs, spatially overlapping NPCs and noisy detections (Fig. 4f, g) This allows direct quantification of geometric features offering insights into spatial ordering and geometry on individual NPC, that are otherwise convolved by overlapping structures or noise. Using the morphological fingerprint as a post-processing step to isolate individual NPC we find the diameter of NPC to be $114 \pm 18$ nm (N = 213) consistent with earlier reports[61] showcasing the utility of SEMORE in extracting protein assemblies and enabling analysis.

The summary of these results on a diverse set of experimental systems demonstrates the ability of SEMORE to generalize to completely different sets of biological systems, imaging and experimental conditions, noise types, and molecular scales without any a priori knowledge of their structure.

## Discussion

The advent, and widespread application, of SMLM, have enabled the imaging of cellular structures with a level of detail that was previously unattainable with live cell imaging. Despite the exponential growth of SMLM imaging techniques, the analysis of the acquired data often relies on manual curation and system-specific analysis, a resource and time expensive process relying on domain expertise. These analytical challenges limit this powerful tool in the determination of protein assemblies' morphology and kinetics impeding our understanding of both the self-assembly mechanisms and, consequently, the physiological responses of these structures especially in systems with no a priori knowledge.

Here we introduce and validate SEMORE an automatic toolbox for morphological fingerprinting to dissect and classify diverse morphologies of protein assemblies achieved from SMLM. SEMORE was designed as a modular, agnostic and unsupervised toolbox to assist analysis across biological systems and experimental configurations unlocking mechanistic insight and unbiased kinetic pathway classification. SEMORE consists of two modules, one for time-aware structural extraction and the second for fingerprint calculations. These two modules are designed to be used either as a general pipeline, or separately for any given purpose. Using a broad set of features maximizes applicability across biological systems and feature ranking can reveal the features of greatest importance providing key mechanistic insights into the given system. SEMORE is currently optimized for laterally stable assemblies, albeit the temporal refinement does handle lateral movement below the agnostic, data-derived search range defined per assembly basis (see Methods). Future versions are planned to include our diffusional analysis framework for motion-aware clustering analysis. Blinking has insignificant effect on SEMORE analysis, however, we recommend common aberrations such as blinking and vibrational or spherical aberrations to be corrected prior to use of SEMORE to avoid any potential misinterpretation of the data. We envision SEMORE's application across diverse systems, the continuous extension of the fingerprinting module and the generation of libraries of protein assembly morphologies. Libraries of morphological features could aid mapping of assembly characteristics to their identity and function for the advancement of biological understanding, and statistical approaches and be the basis for future supervised learning purposes (Supplementary Fig. 13).

SEMORE directly predicts on experimental data circumventing the need for large manually annotated data sets and ensures its widespread applicability across systems without the need for retraining or of a priori knowledge. In essence, the only human intervention in transiting from raw x, y, (t) coordinates of biomolecular assemblies to outputting their diverse morphologies is selecting the initial parameters of the DBSCAN or HDBSCAN. We demonstrate this operational utility on five diverse experimental data sets: temporarily resolved protein aggregation by REPLOM, static nuclear pore complex and fibroblast growth receptor 1 (supplementary Fig. 17) imaging by dSTORM as well as two experimental data obtained from dynamic, live-cell PALM (Supplementary Figs. 18 & 19), as well as multiple ground truth simulated data sets. In all cases, relevant structures are precisely extracted and featured. Effectively, SEMORE operates across five experimental data sets, with spatial dimensions spanning 3 orders of magnitude, from nanometers[58] to micrometers[19], and temporal dimensions spanning from milliseconds[60] to seconds[19]. This fact provides strong support for SEMORE as a universal, input-independent model for the SMLM community to use in conjunction with, or to be incorporated into SMAP, or as a convenient standalone toolbox.

Current advances in live-cell and live-tissue imaging[44,52,62], optical imaging techniques allow tissue-scale 4D cell biology[33,39,41], but currently, available analytical pipelines are inadequate for automatic and agnostic analysis of the temporal evolution of high-content protein clustering imaging data. SEMORE is, to the best of our knowledge, the first toolbox that automatically and agnostically analyses and classifies temporal evolution of morphology changes of protein clustering in super-resolution images, addressing a significant bottleneck of the field. SEMORE can serve as a platform accommodating the wealth of 4D cell data of e.g. coated pits formation, synuclein fibrillation in cells, protein phase separation, and actin filament elongation, to mention a few[20,22,56]. We anticipate the methodology to be directly applicable across biological systems, imaging and experimental conditions, or laboratories, to dissect diverse types of static or dynamic protein assemblies resolved by super-resolution imaging.

## Methods
### Mathematical definitions
For all distance calculations both in real and parameter space, unless different is stated, Euclidean distance is used and defined as

$$d(p,q) = \sqrt{(p_1 - q_1)^2 + (p_2 - q_2)^2 + \ldots + (p_n - q_n)^2} \tag{1}$$

**Statistical distributions** are calculated as follows:

$$\text{Gaussian distribution}: N(x) = \frac{1}{\sigma\sqrt{2\pi}} \cdot e^{-\frac{1}{2}\left(\frac{x-\mu}{\sigma}\right)^2} \tag{2}$$

$$\text{Poisson distribution}: P(k) = \frac{\lambda^k e^{-\lambda}}{k!} \tag{3}$$

For **data transformation**, both Standardized and MinMax are used throughout the SEMORE pipeline and are performed as follows:

$$\text{Standardized}: \hat{x} = \frac{x - \mu}{\sigma_x} \tag{4}$$

$$\text{MinMax scaler}: \hat{x} = \frac{x - x_{\min}}{(x_{\max} - x_{\min})} \tag{5}$$

**Performance metrics** are used to evaluate the performance of different classification tasks.

For the task of summarizing the classification performance different aggregation approaches are used for completeness or chosen based on applicability: Micro) Accumulating all outcomes for each individual entry and calculating a total statistic. Macro) Calculating the desired performance metric on a label basis followed by averaging

across labels. Weighted) Calculating the performance metric as in Macro, but weighing each entry by the total number of TP. To restrain a possible bias in reported classification evaluation metrics towards larger aggregates, the macro average is generally used unless otherwise stated. The accuracy of the clustering segmentation is defined as:

$$Accuracy = \frac{TP}{(TP + FP + FN)} \quad (6)$$

for each aggregate resulting in a performance metric based on correctly annotated localizations, disregarding the relative size of the cluster. TP: True positive, FN: False negative (when the aggregate data points have been predicted as noise) and FP: False positive (When an aggregating point is predicted as another aggregate, happens when two independent aggregates are predicted for the same ground truth aggregate). As this is treated on an aggregate basis, there would be no TN to conclude. For general performance, the Precision, Recall and F1 score are calculated as:

$$Precision = \frac{TP}{TP + FP} \quad (7)$$

$$Recall = \frac{TP}{TP + FN} \quad (8)$$

$$F1 = 2 \cdot \frac{Precision \cdot Recall}{Precision + Recall} \quad (9)$$

## SEMORE clustering module

The presented pipeline of SEMORE's clustering module of temporally resolved structures is as follows:

1. **Standardization** of spatiotemporal localizations by z-score in all 3 dimensions (x, y, t) to ensure equal distance measure and density impact effectively ensuring scale invariance for the clustering module.
2. **Initial clustering** by either Density-Based Spatial Clustering of Applications with Noise (DBSCAN) or High Density-Based Spatial Clustering of Applications with Noise (HDBSCAN) or. The model is chosen agnostically based on the general density of the field of view (FOV) where for less than 1500 data points per standardized area the DBSCAN is used. The primary objective of the initial clustering is to identify regions of interest with significant probability of containing a target structure or potentially containing overlapping structures. It is not necessary to have precise clustering as later fine-graining and temporal refinement improves the segmentation, but more accurate initial segmentation can result in an overall better performance.
3. **Topological failsafe** to prevent nonsensical clustering in experimental data. It is assumed that there is at least one region of interest in a given FOV, but if the initial clustering model does not find any regions, a density analysis is performed to identify the best suggestion for a high-density area, thus refining and improving the initial clustering. The density analysis is done through a Gaussian blur applied to the 2D binned locations, and the biggest contour region containing the 90th percentile of the density is chosen to be the only initial cluster.
4. **Temporal refinement and semi-supervised hyperparameter choices** is then done for all initially found regions of interest with the goal of dissecting whether identified regions are single clusters or individual clusters grown to overlap over time. The regions are "re-scaled" through a MinMax scaler to capture all locations and achieve comparable scaling in the time axis compared to the Euclidean space. For each region, a region-specific search range is calculated from the pairwise distance of all data points in the

region. This search range is defined as the square root of the 95% confidence interval of the standard error on the mean for the interquartile distances.

$$dist_{investigated} \in IQR(pdist) \quad (10)$$

$$r_{search} = \sqrt{SEM \cdot 1.96} = \sqrt{95\%CI_{SEM}} \quad (11)$$

The search range value is used both in the search for new aggregational seeds as well as linking localizations to existing aggregates signifying growth. The temporal refinement is done in a frame iterative manner, from lowest to highest, for each frame the corresponding data points accumulate as points for investigation. All data points for investigation are initially label-free and get labeled assigned in the following way: For each step, a DBSCAN (eps = search range, min_sample = 50) is fitted on the label-free points of investigation for detection of new aggregation seeds, while a radius-based nearest neighbor (r = search range) model is fitted on the already labeled points and predicts on label-free points for the investigation to classify further growth of existing assemblies. Data points not assigned are still label-free in the next step until all frames have been investigated, in which any data points left unlabeled are considered noise.

5. **Smart density filter** to ensure that the SEMORE clustering module extracts only actual localizations and not noisy detections, a density filter is applied, based on the assumption and observation that aggregates have a higher density than noise. The noise density is estimated in the intermediate step, between the first and second steps of the clustering module, by randomly selecting 500 data points from the initially found noise and calculating the density of each point. To avoid underestimation induced by the empty space from the high-density areas, only densities above the 25th percentile is used to calculate the mean and standard deviation. An identified structure must have a density of one standard deviation plus the average density to be extracted as an actual structure by temporal refinement, otherwise, it is classified as a noisy detection. Furthermore, a minimum data-point requirement is also added, which is included as a "and" statement for the filter (see Supplementary Fig. 1).

## SEMORE Morphology fingerprinting

The fingerprinting module consists of 40+ total features deriving from four main feature classes: 1) Symmetry, 2) Geometric, 3) graph network and 4) circularity, the description for each feature is found in Supplementary Table 1. All features are computed directly on clusters identified by the clustering module but works directly on any set of localizations. To ensure consistency in feature computation, all identified structures have their localizations standardized and projected, so the major axis points in the same direction across structures.

**Symmetry.** For calculation of the symmetry **(7 features)**, the center of mass is subtracted from the positions centering all points around the origin. The symmetry features of localizations can be computed based on counts per quadrant.

**Geometric.** For the geometric **(3 features)**, a Delaunay triangulation is performed on all localizations in a given structure, from which triangle edge distances are used to fit a lognormal distribution, pruning any identified triangle edges with less than a corresponding 5% right-tailed probability with the remaining edges allow for accurate and computationally effective area estimation as seen in Supplementary Fig. 20.

**Graph network.** The graph network **(25+ features)** is initiated through a radius-to-neighbors graph with the radius as the distance threshold

obtained from the Geometric feature class. From the resulting graph, a minimum spanning tree is constructed using the length of connections as "weights".

**Circularity.** Calculation for the **Circularity (5 features)** feature class is based on a contour line around the highest density area in the aggregate identified by a two-dimensional histogram. Specifically, the 2D histogram counts are Gaussian-blurred followed by a binary threshold filter at the 90-percentile producing a set of border coordinates (see Supplementary Fig. 21).

**Hyperparameters of SEMORE**
The general hyperparameters of SEMORE clustering module consist of the native parameters in the initial clustering module (DBSCAN or HDBSCAN) which as default, in this work, are set to values identified empirically as broadly well-performing for capturing areas of interest in data containing localizations produced by protein aggregation (as presented in this article). It is important to note that the agnostic parameters used for temporal refinement and output filtering are dependent on calculations made from the high-density areas, they are robust and small permutations will not have a significant effect. However, extreme over-clustering in the initial step can result in faulty output. Additional hyperparameters, while less likely to need adjustments, are *investigate_min_sample* as well as *radius_ratio* which by default are set to 50 and 1.96 respectively. These parameters are used in the temporal refinement, *investigate_min_sample* is used for the DBSCAN to find aggregation start seeds while *radius_ratio* is used in the general search range for both DBSCAN and aggregate growth. *investigate_min_sample* should only be changed if an incorrect number of underlying clusters are generally initiated through the temporal refinement while *radius_ratio* changes the growth extends each frame, this should only be adjusted if underlying aggregates are mainly one-sided dominated i.e., the first initiated aggregate engulfs most of the high-density area. The last parameters are *rough_min_points*, *final_min_points* and *filter_mode* which all affect the filtering of aggregates between the first and second layers with *rough_min_points* filtering to small aggregates before the temporal refinement, while the *final_min_points* does the same after the temporal refinement, and filtering type set to strict, lose or none, which control if an aggregate has to pass both the density and point amount filtering, only one of them or none of them. In general, SEMORE clustering achieves high performance with out-of-box settings as seen in (Supplementary Figs. 2–4). For all default values of hyperparameters, we direct the reader to GitHub.

**Simulations and treatment**
The simulation is based on typical structures observed in protein aggregation experimental data and is created in a frame-iterative manner within a 40 µm x 40 µm field of view. Aggregates are initialized from a randomly drawn or given starting point within the first 300 frames, with a minimum aggregation time of 100 frames and a maximum experiment time of 400 frames. The number of points added per frame is randomly drawn from a type-specific distribution, all with the possibility of adding zero.

**Isotropic growth.** The symmetric isotropic aggregates are simulated using a Gaussian Kernel Density Estimator (KDE) that is fitted to the available points, with the bandwidth increasing as a function of the growth time:

$$bandwidth = ((frame - frame_{start})/(frame_{end} - frame_{start})*frame_{end} + 1)*10 \quad (12)$$

Next, a uniformly drawn number of points (between 0 and 30) is sampled from the fitted KDE and added to the total number of points in the aggregate. The KDE is then refitted to the updated set of points,

resulting in a symmetrical circular aggregate that increases in size over time.

**Sterically hindrance (random).** The steric hindrance growth simulation resembles a Monte Carlo method, where the underlying probability of growing a specific branch, depends on the hindrance in that area. This is a more natural, stochastic growth pattern which we dubbed "random aggregate". In this approach, for each of the last 50 added points (or what is available if less), a candidate for a new point is drawn from a unit Gaussian distribution around the point. For each candidate, the distance is calculated to 50 or fewer current points. A measure of steric hindrance is then calculated using the formula:

$$hindrance = \sum \exp(-distance^2) \quad (13)$$

A Poisson-drawn number (average 10) of points with the lowest hindrance is then selected and added to the set of points. Drawing from the sterically affected underlying distribution results in increased branching and heterogeneous aggregate structures compared to the isotropic simulations.

**Fibril growth.** To simulate the thin and branching structures commonly observed in fibril aggregates, a directed diffusion scheme was used with a starting direction drawn from a uniform distribution (between 0 and 2π). At each step, 3 major features must be drawn: the number of points to be added, new direction and distance. The number of points to be added is determined by a Poisson distribution with an average of 1. Direction and distance are both drawn from a Gaussian distribution, to depict a directed rod-like structure the new direction has the previous direction as the mean and a standard deviation of π/4, while the distance is simply having a mean of 100 nm and a standard deviation of 20 nm. Furthermore, for each step, there is a low probability of the fibril branching out. The angle of the branching is also drawn from another Gaussian distribution with the previous direction +/- π/4 as the mean with π/16 as the standard deviation, the plus or minus indicates which side of the fibril the branch is sprouting from and is determined from a Bernoulli trial with p = 50%. For the simulations used, the probability of branching was set to 0.5% with a maximum number of branches set to 3.

**Sparse aggregate simulations.** Small, sparse aggregates were simulated for the Supplementary Figs. (7–9) to evaluate the effect of number of detections on both the clustering module and morphologically fingerprinting. These structures were inspired by nanostructures and simulated in a 3 µm x 3 µm region of interest. Temporally resolved fibril and static elliptical structures were simulated for 4, 8, 15 and 25 points. For each simulated experiment 30 protein assemblies were grown equally distributed between the two morphology types and noise was applied either uniformly or heterogeneously (see respective method sections).

**Sparse fibril simulation.** The temporally resolved fibril structures were created as per the described Fibril growth in methods with a mean elongation distance of 20 and sigma of 5.

**Sparse elliptical structure simulations.** The non-temporally resolved ellipse structures were inspired by Nieves et al[35] and created using two Gaussian distributions with sigmas of 10 and 20 to draw x and y values respectively. In addition, random rotations were applied to the finished structure. Randomly drawn individual frames were assigned to each localization in these structures stochastically, representing SMLM data obtained from conventional static methods.

**Blinking noise from ground truth.** Blinking noise-induced structures were produced through inspiration from Nieves et al[35]. A ground truth

structure was simulated as either fibril or elliptic as seen in methods. For each position in the ground truth structure 1–6 detections were uniformly drawn. Centered on the ground truth position these additional detections were displaced by a localization error drawn from a Gaussian distribution with a unique sigma drawn from a lognormal distribution with a mean of 3 and a sigma of 0.28 as per Nieves et al.[35]. The original ground truth detections are not included in the final structure.

**Treatment of simulated data.** For all protein assembly segmentation in simulated studies, the hyperparameter *investigatet_min_sample* is changed from the default 50 to 25 lowering the requirement of aggregate seed detection. This is possible because, in real experiments, noise can be induced by small non-aggregating clumps of docking which are mistakenly seen as growth seeds. This has been counter-measured by the default settings of SEMORE temporal refinement hyperparameters. However, simulated studies have a controlled and perfectly uniform distributed noise, removing these clumps and therefore lowering the requirement of aggregate seed. All hyperparameters remain default throughout the aggregate dissection unless different is stated. For DBSCAN and HDBSCAN comparison, hyperparameters were optimized for a representative image for each simulation and used throughout the analysis.

For the **segmentation of the symmetric isotropic** aggregates, the SEMORE was initialized with the HDBSCAN hyperparameter *cluster_selection_epsilon* changed from 0.03 to 0.05. This was done for the initial clustering to capture the edges of the isotropic aggregates. This part has a drastically lower density than the middle of the aggregate, the default values of the HDBSCAN would not have included it in the initial clustering. The **random** aggregates were segmented without any further changes to the hyperparameters.

The highest amount of hyperparameter alterations was required by the **fibril** segmentation. Due to the smaller spatial size, lower density, and rod-like structure of these aggregates, adjustments were necessary for both the initial clustering and temporal refinement processes. Specifically, the HDBSCAN used in the initial clustering had to be tuned to detect smaller aggregates and areas, *min_cluster_size was* changed to 60, *min_samples* to 30 and *clustering_selection_epsilon* was set to 0 these settings allowed for the initial clustering to capture the aggregates while minimizing additional noise inclusion. For the temporal refinement to properly dissect the fibril from either overlap or noise, the *radius_ratio* was lowered to 1, as this allows for a smaller search range, and therefore includes minimal noise.

The out-of-box SEMORE fingerprint module was directly applied to all the segmented aggregates, the unique morphology fingerprint was concatenated into one large dataset, blinding the original simulation origin. The accumulated fingerprint was reduced in dimensionality through a 3-component UMAP (n_neighbors = 5, min_dist = 0.1) revealing the 4 prominent clusters. Additionally, the circularity feature class of the fibril aggregates were treated independently through the same UMAP, to reveal the morphology diversity from branching to non-branching fibrils, which were thoroughly performance tested (see Supplementary Fig. 13).

**Sparse aggregate treatment.** While the general low density of the simulations, will cause SEMORE to select DBSCAN as the initial clustering model, minor changes to the min_sample hyperparameter were introduced to improve segmentation of the sparse aggregates, the change was min_sample = 10, 7, 5, 3 for 25-, 15-, 8- and 4-point settings respectively, only additional change was to requirement of final aggregates sizes to be larger than 5 points.

For all sparse structures, the morphology fingerprinting (excluding "N_points" to counter any correlation between segmented size of noise detections and actual protein assemblies being used classification) was extracted and used to produce 2-component UMAP

representations for visualizing the structural information contained within the morphological fingerprint.

**Induced blinking noise structure treatment.** The SEMORE fingerprinting module was applied directly to the simulated structures without the SEMORE clustering module, due to its static nature, and the aim of investigating the fingerprint's robustness to perturbations. The fingerprints of the ground truth structures were extracted and visually compared to their blinking-noise counterparts through an out-of-box 2-component UMAP embedding. Each structure was embedded individually and coloured-coded correspondingly to the morphology class type to enable a fair and transparent visual comparison.

**Creation and treatment of dynamical changing morphology.** To simulate a protein assembly with dynamic morphology in which morphology class transitions within classes and in between classes of; fibril, isotropic and sterically hindered asymmetric, multiple protein assemblies were simulated for each morphology class. 125 protein assemblies were created in 25 intervals, the 3 types of morphologies were equally distributed and pruned to a maximum of 200 points. Each structure was sorted with respect to its morphology class and appended to the final temporally evolving structure, with each structure being a new temporal morphology state. This results in a rapidly changing morphology where each temporal state was embedded using a 2-component UMAP.

**Heterogeneous noise generation by perturbation and noise seeds.** Non-uniform noise was generated by placing 5–25 noise seeds uniformly in the field of view. Each noise seed contributes 20–50 points Gaussian from their noise seed origin with a sigma of 320 nm. To further reduce uniformity all noise points drawn from these Gaussian noise seeds are further displaced randomly by a Gaussian with mean 20000 nm and sigma 100 nm distribution.

### Experimental data and treatment

**Real-time observations of insulin aggregation using REPLOM.** 10 data sets containing locations obtained from experiments containing human insulin amyloid-spherulite aggregates were provided and produced through REPLOM as seen in Zhang et al.[19]. Each data set was treated through both SEMORE out-of-box clustering and fingerprinting. For Isotropic and anisotropic separations, the circularity features alongside L_s_ratio and L_l_ratio were used for the UMAP (min_dist = 0.1, n_neighbors = 10) embedding, from which a DBSCAN (eps = 0.7, min_sample = 5) were used to classify the two presented clusters in the UMAP-embedding and visualized along its circularity estimate for Fig. 4.

**Nuclear pore complex imaged by dSTORM.** Images containing dSTORM assay of nuclear pore complexes were downloaded from https://srm.epfl.ch/ dataset page original from Li et al.[58], the locations were extracted using simple in-house detection software with no correction for x,y-drift as only the first 20,000 (out of 150,000) frames were used to achieved suitable amount locations to perform the required analysis. The locations were up scaled by multiplying x and y 132 nm/pixel and 142 nm/pixel respectively from where only a subset of the locations contained in the middle of the experiment was investigated (Sub-set rectangle mask drawn by [5000 nm, 5000 nm] to [12,500 nm, 12,500 nm] corners). The locations were clustered by DBSCAN (eps = 50, min_sample = 20) which resulted in 283 clusters. The morphology fingerprint was extracted for all found clusters by the SEMORE fingerprint module without Gaussian fits the underlying density distributions (mu_N,sig_N and W_N). All features were used in the UMAP (min_dist = 0.2, n_neighbors = 10) embedding from which the 3 presented clusters were extracted using a DBSCAN (eps = 0.8, min_sample = 5). The "Area" feature was extracted from embedding

cluster "1" and used to calculate the radius assuming a circle: $r = \sqrt{\frac{A}{\pi}}$. The radius distribution was fitted with a Gaussian distribution through maximum likelihood from which both mean and standard deviation was found. The diameter error was found through error propagation as $\sigma_d = \sqrt{4\sigma_r^2}$.

## Statistics & reproducibility

This study includes no statistical test and no data were excluded. SEMORE works directly on data without supervised learning, thus always blind to test data, and model evaluation was performed on multiple diverse experimental systems, as well as, simulations chosen to resemble experimental observations. These test sets provide biologically relevant sample sizes and variance. All data and scripts are publically available for reproducibility. The experiments were not randomized before evaluation.

## Reporting summary

Further information on research design is available in the Nature Portfolio Reporting Summary linked to this article.

## Data availability

The data used in this study are available on GitHub under https://github.com/hatzakislab/SEMORE and the electronic research database of University of Copenhagen with https://doi.org/10.17894/ucph.7f5ea282-faa5-4519-987d-13df11073a7b. In addition, we refer to the original publications of the data: Zhang et al.[19], Nieves et al.[35], Li et al.[58], Hou et al.[59] and Wallis et al.[60]. We thank Hou et al.[59] for generously donating data. Source data are provided with this paper.

## Code availability

All code is written in python, and can be accessed on GitHub: https://github.com/hatzakislab/SEMORE, https://doi.org/10.5281/zenodo.10371199.

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

## Acknowledgements

This work was supported by the Villum foundation center BIONEC (18333), the NNF challenge center for Optimised Oligo Escape and control of disease (NNF23OC0081287), the NNF center for 4D cellular dynamics (NNF22OC0075851), the Lundbeck foundation grant R346-2020-1759, Villum foundation Synergy grant (40578), and Carlsberg foundation grant CF21-0499. We are grateful to the group of Professor William Edward Louch for providing the experimental data sets of live-cell PALM of ryanodine receptors. N.S.H. is a member of the Integrative Structural Biology Cluster (ISBUC) at the University of Copenhagen and associate member of the Novo Nordisk Foundation Center for Protein Research, which is supported financially by the Novo Nordisk Foundation (NNF14CC0001).

## Author contributions

J.K.H. designed the research project with input from NSH, S.W.B.B. performed all computational analysis, developed the SEMORE model supervised by J.K.H. and N.S.H. S.W.B.B. developed simulations assisted by M.W.D. and M.Z. provided data. S.W.B.B., J.K.H. and N.S.H. wrote the paper with feedback from all the authors. N.S.H. with major contribution from JKH, had the overall project management.

## Competing interests

The authors declare no competing interests.
