## [Peer Review File · Nature Communications]

SEMORE: SEgmentation and MORphological fingErprinting by machine learning automates super-resolution data analysisReviewers' comments:

Reviewer #1 (Remarks to the Author)

Bender et al., present a machine learning based clustering and classification algorithm, SEMORE, for the segmentation of diverse cluster morphologies from heterogenous SMLM data. Being able to identify and segment clusters in heterogenous data is a key problem in SMLM, so the algorithm is a well-placed idea for addressing this. The algorithm is initially demonstrated on simulated data, consisting of 3 biologically relevant clustering scenarios; symmetric clusters, asymmetric clusters (non-equal growth) and fibre-like structures. The algorithm is then demonstrated on real experimental SMLM data, namely insulin aggregation and nuclear pore complex data. In the case of the insulin aggregation, with challenges in segmentation were overcome using temporal information. The paper addresses two important problems with clustering of SMLM data in general; the need for user inputs to define the segmentation, and post-analysis classification of segmented clusters to understand biological phenomena of protein distributions and assembly. The manuscript is well presented and easy to follow. Furthermore, the SEMORE software is easy to find with clear installation instructions. However, I found it slightly challenging to get running in a new environment (admittedly I have a basic working knowledge of python, but see below comment). Additionally, the data from the publication was all provided, and easy to look through, which should be highly commended. There are a couple of conceptual and practical points, however, that I think need addressing, which would strengthen the manuscript.

I would recommend publication of the manuscript if the comments below can be addressed.

Major comments:

- The clusters within the simulated data seem to be very large in scale, some on the order of 10 microns. This is likely to be an uncommon scenario in biological SMLM data, especially when it comes to clustering of that data. I would recommend some simulations on the scale of the NPC data, i.e., several clusters sub-micron with diverse morphology. A recent paper cited in the manuscript proposed several different clustering scenarios at this scale (Nieves et al., NatMeth, 20, pages 259–267, 2023) where two different cluster types are present within the region. This would be really nice to see clusters of similar density (in that paper it was approximately 10-20 per 3x3 micron²), but analysed in a similar fashion here, i.e., analysis of the whole 40x40 micron region. This would be quite powerful if whole fields of view could be analysed quickly, without the need for subdivision of the data (see later point on performance).
- It would be interesting to know a little more about the limits of SEMORE, as in biological SMLM data there may be several different classes of cluster, which may have stronger similarities, within the same ROI. For example, if a protein is present as a monomer and can assemble into higher order oligomers, it might be expected there are some low-level oligomers (4 proteins clustered proteins at sub-diffraction, e.g., tetramers) in the data at time of fixation. Would these small clusters be distinguishable from the noise class? Some examples of how SEMORE deals with this low-level oligomerisation (low-density clusters) would strengthen the manuscript.
- Following on from this point, one thing lacking from the manuscript is any comment on a common problem for arguably the two most common SMLM techniques (PALM, and dSTORM - used here); multiple blinking of the fluorophore. Normally, this introduces artefactual clustering into the SMLM data, and several approaches now exist for correcting this (e.g., Jensen et al., NatMeth, 19, pages 594–602, 2022, and Bohrer et al., NatMeth, 18, pages 669-677, 2021). Thus, in relation to biological data, some structures that may appear clustered, and then segmented in uncorrected data, will not be so after correction, shifting the observed distributions of the classes. Would the authors recommend such corrections before use of SEMORE for accurate fingerprinting? Also, some sense of how the lack of blink correction could impact classification would be beneficial. This could be done by comparing a ground truth of the molecule positions and a version where the data has added blinking.

Minor comments:

- All figures showing SMLM data or plots of clusters etc. need a scale bar or some axes. It was hard to appreciate, until using the code, how different the length scale of the clusters between simulations and experimental data were (some several microns, whilst NPC data is much smaller scale)

- I ran the SEMORE installation as directed on the GitHub, generating a new environment. I found quickly that I was missing the dependencies needed to run the scripts (e.g., opencv in the dependency file should be opencv-python, I assume this is why this file may not have worked for me). Further, other more basic packages are of course not there in a new environment, e.g., pandas, numpy, etc., and require the user to add them. I would recommend the authors give a full and correct dependencies file, as this troubleshooting would be beyond many of the potential users with limited coding experience.

Reviewer #2 (Remarks to the Author)

Bender et al. present SEMORE, SEgmentation and MORphological fingErprinting, as a semi-automatic machine learning framework for the analysis of super-resolution data to extract and quantify protein structures. By implementing a multi-layered density-based clustering module and a morphology fingerprinting module for quantification, they demonstrated SEMORE on simulations and experimental super-resolution data, including insulin aggregates and nuclear pore complexes. Developing automatic clustering analysis without a priori knowledge and tracking the temporal evolution of morphology changes of protein clustering from single-molecule localization microscopy image data is certainly a good direction for the development of super-resolution microscopy field. However, in my view, their demonstrations are limited to the analysis of the growth of immobile clusters under well-controlled noise conditions, which could be quite different from general biological behaviors. This raises concerns about the limited applicability of this method. Additionally, although the authors insist that their method represents the first time-aware framework for 4D super-resolution data, they mainly focus on presenting the classification results for protein cluster growth, without showing identified cluster images at each time point. Therefore, it is hard to judge whether their method can effectively capture the temporal changes of protein clusters. To claim that their method is genuinely powerful for automatic clustering analysis and tracking the temporal evolution of protein clusters, they should also show the time-resolved clustering data.

Furthermore, I cannot see the significant novelty of this method; they mainly used the reported clustering method (DBSCAN) for cluster identification. Although it appears that they additionally conducted temporal refinement after the DBSCAN analysis, this additional step seems to be mainly for dissecting clusters over time, rather than advancing the clustering analysis method. The new development of their method seems to be a morphology fingerprinting module that uses multiple feature vectors for quantification. However, the performance of this method for real experimental time-resolved super-resolution image data was demonstrated only for the relatively new imaging method REPLOM, which is not a general type of data. To demonstrate the general performance of this method, the authors should present more general time-resolved single-molecule localization microscopy data, rather than static dSTORM data. Although the authors applied their method to NPC data, it appears to be just static data, and the categorization of the clusters is not biologically meaningful (individual, overlapped, noise). For these reasons, I believe the authors' claims are not well supported, which unfortunately does not warrant publication in Nature Communications in the current form of the manuscript.

1. As suggested above, the main problem with the current manuscript is the lack of time-resolved analyzed data. They only showed the classification results from simulation data for isotropic, random, and fibril growth. However, the time-resolved cluster morphological change data (at each time point) should also be presented to assess whether their method correctly identified each growth pattern. Additionally, I would like to see the time-scaled morphological changes of insulin aggregates and nuclear pore complexes from experimental data to evaluate whether their method performs well even with real data under heterogeneous noise conditions.

2. The demonstrated data seems to only focus on the growth of clustering. However, changes in protein clusters in the real world could involve not only cluster growth but also significant morphological changes (e.g., spherical to tubular), shrinkage, movement, and scattering (depolymerization). For wider applicability of this method, the authors should include related data as well.

3. Their demonstration is mostly based on simulation data, where the noise appears to be well controlled and homogeneous. Although they tested this method under different noise density conditions, it seems that the density they used is uniformly distributed across the entire field of view and throughout the entire time domain, which could differ from real conditions. Additionally, the intensity of noise could be heterogeneous. Because such heterogeneity of noise could affect the identification and classification of protein clusters, they should test the method with various types of noise.

4. The authors claim that the use of a machine learning-based method is an important aspect of this study. However, the machine learning method employed in this paper appears to be limited to unsupervised point clustering, rather than classification or other methods. Moreover, these unsupervised point clustering methods used here do not seem to differ significantly in terms of methodology from the existing method, DBSCAN. I believe the authors have not sufficiently explained why the method they used can be considered a machine learning approach and what sets it apart as a novel contribution compared to existing methods.

5. In the Fingerprinting module, it is mentioned that over 40 features were utilized to analyze the properties of each cluster using various methods. However, it appears that a comprehensive assessment of the strengths and weaknesses of these methods was not conducted. It is unclear why the given methods are expected to yield good features for cluster classification. Additionally, it would be beneficial to determine if all the provided methods are necessary based on the analysis results. For instance, Supplementary Figure 8 suggests that it might be possible to identify a better method instead of using all the feature vectors obtained from each feature. There is a possibility that some of them could even interfere with the analysis process, so it would be insightful to see further results in this regard.

6. In general, the methodology is not thoroughly explained. For instance, even if the method applied in the existing DBSCAN is used as is, it would be helpful if the methodology were described in more detail. Additionally, the methodology for the classification of each point, which occurs after the fingerprinting module and embedding with UMAP, is not elaborated very well. Since the fingerprinting module appears to be the only distinctive novelty of this method, as mentioned above, it should be described in detail to provide a clearer understanding.

Reviewer #3 (Remarks to the Author)

The manuscript entitled "SEMORE: SEgmentation and MORphological fingErprinting by machine learning automates super-resolution data analysis" by Hatzakis et al introduces SEMORE, a semi-automatic machine learning framework for universal, system and input-dependent, analysis of super-resolution data. The manuscript is well written and informative. However, this manuscript can be improved by addressing the following issues:

1. The authors have developed a new analytical tool SEMORE to extract and quantify underlying structures limiting single-molecule localization microscopy (SMLM). How this newly developed tool SEMORE is better suited with SMLM than some of the previous employed tools such as SMAP (modular super-resolution microscopy analysis)

2. The authors should explain how SEMORE deals with commonly occurring supervised and unsupervised machine learning problems.

3. The authors should also highlight the role of SEMORE in advancement of machine learning-based approaches.

4. Since vibrational and spherical aberrations prove to be greater hindrance at high resolution. Moreover, live samples are more adversely affected by super-resolution imaging because of high excitation intensity or extended exposure times. Does the SEMORE tool take these concerns into consideration?

Reviewers response

Reviewer 1	2
General response	2
Comment 1.1	2
Comment 1.2	4
Comment 1.3	5
Minor comments:	6
Reviewer 2	6
General response	7
Comment 2.1	9
Comment 2.2	12
Comment 2.3	14
Comment 2.4	14
Comment 2.5	17
Comment 2.6	18
Reviewer 3	19
General response	19
Comment 3.1	20
Comment 3.2	20
Comment 3.3	22
Comment 3.4	23
Figures	25
Response Figure 1: dSTORM data from Nieves et. al. 1 clustered and quantified by SEMORE	25
Response figure 2: Sx1a-mEos2 sptPALM data from Wallis et. al. 3 clustered and temporally quantified by SEMORE.	26
Response Figure 4 Evaluation of SEMORE on temporarily resolved live-cell PALM data of ryanodine receptors (RyRs).	27
Response Figure 4: Evaluation of SEMORE' segmentation and structural information extraction on simulated small tetramer assemblies.	28
Response figure 5: SEMORE classification performance for small, sparse clusters of diverse morphologies.	29
Response Figure 6: Deconvolution of temporal refinement on real and simulated data.	31
Response Figure 7: Visual representation of SEMORE's ability to capture morphological growth in time	32
Response Figure 8: Depiction of recurrent SEMORE fingerprinting for dynamic morphology variation of protein clusters.	33
Response Figure 9: Benchmarking SEMORE on for heterogeneous, non-uniform noise.	34
Response Figure 10: Effect of blinking on SEMORE's morphological fingerprinting and characterization of morphological classes.	35
Response Figure 11: Demonstration SEMORE clustering of morphology shrinkage	35

Reviewer 1

Bender et al., present a machine learning based clustering and classification algorithm, SEMORE, for the segmentation of diverse cluster morphologies from heterogenous SMLM data. Being able to identify and segment clusters in heterogenous data is a key problem in SMLM, so the algorithm is a well-placed idea for addressing this. The algorithm is initially demonstrated on simulated data, consisting of 3 biologically relevant clustering scenarios; symmetric clusters, asymmetric clusters (non-equal growth) and fibre-like structures. The algorithm is then demonstrated on real experimental SMLM data, namely insulin aggregation and nuclear pore complex data. In the case of the insulin aggregation, with challenges in segmentation were overcome using temporal information. The paper addresses two important problems with clustering of SMLM data in general; the need for user inputs to define the segmentation, and post-analysis classification of segmented clusters to understand biological phenomena of protein distributions and assembly.

The manuscript is well presented and easy to follow. Furthermore, the SEMORE software is easy to find with clear installation instructions. However, I found it slightly challenging to get running in a new environment (admittedly I have a basic working knowledge of python, but see below comment). Additionally, the data from the publication was all provided, and easy to look through, which should be highly commended. There are a couple of conceptual and practical points, however, that I think need addressing, which would strengthen the manuscript.

I would recommend publication of the manuscript if the comments below can be addressed.

General response

We are grateful for the favorable and critical comments, for acknowledging the manuscript easy to follow and for recommending publication after the comments are addressed. Fully and in detail addressing the “conceptual and practical” comments helped us further improve the manuscript and the quality and user-friendliness of our open-source implementation on Github. Figures created following your and other reviewers comments have been pasted below for your convenience.

Comment 1.1

- The clusters within the simulated data seem to be very large in scale, some on the order of 10 microns. This is likely to be an uncommon scenario in biological SMLM data, especially when it comes to clustering of that data. I would recommend some simulations on the scale of the NPC data, i.e., several clusters sub-micron with diverse morphology. A recent paper cited in the manuscript proposed several different clustering scenarios at this scale (Nieves et al., NatMeth, 20, pages 259–267, 2023) where two different cluster types are present within the region. This would be really nice to see clusters of similar density (in that paper it was approximately 10-20 per 3x3 micron²), but analysed in a similar fashion here, i.e., analysis of the whole 40x40 micron region. This would be quite powerful if whole fields of view could be analysed quickly, without the need for subdivision of the data (see later point on performance).

Response 1.1

The reviewer correctly points out that large aggregates are generally depicted in the manuscripts. We stress here that SEMORE clustering module is scale-invariant and solely relies on localisations. This is due to our choice of 3D axis standardization. Working with standardized dimensions enables SEMORE to work across experimental configurations and dimensionalities. This allows SEMORE to operate independently of the imaging dimension and achieve high classification accuracies solely based on the number of localizations contained within the protein assembly given the spanning area. The comment still raises a very valuable point on the generalization and performance of SEMORE and to demonstrate this and fully answer the comment of the reviewer we performed two studies:

- 1) Simulated diverse, nanometer dimension structures, and quantified SEMORE performance on their segmentation and classification for data with 8, 15 and 25 localisations in an area that corresponds to $\sim 1\mu\text{m}$. As can be seen in Supplementary Fig. 8 (Response figure 5) in the revised version, SEMORE reached an accuracy of $>90\%$ in extracting clusters with 8 or more localisations with the morphology fingerprinting completely separating true clusters from noise.
- 2) Simulated protein assemblies of just 4 detections and evaluated SEMORE's performance on such a small cluster representing the minimal representation of a tetramer. SEMORE achieves accuracies of up to 70-90% in segmenting these tiny structures depending on the noise levels while the morphological fingerprinting module captures these noisy detections allowing for potential post-processing and accuracy increase, Supplementary Fig. 7 (Response figure 4).
- 3) Evaluated the number of points needed for SEMORE to accurately detect the initiation points of spatially overlapping clusters. The high accuracy of SEMORE to capture initiation points can be quantified and visualized using simulated data showing is just ~ 13 frames off on average across 3 types of spatially overlapping morphologies. Furthermore, qualitative assessment using experimental data of temporally resolved insulin aggregation supports the reliability of SEMORE, Supplementary Fig. 5+6, response figure 6+7).

In essence, ~ 10 detections is enough for SEMORE to obtain high segmentation and classification accuracy which is well below the common number of detections in most SMLM experiments. We analyze dSTORM data of Nieves et al. NatMeth, 20, pages 259–267, 2023¹ obtaining similar densities and cluster sizes but in fact analyzing the entire field of view at once see new SI fig 17 (response figure 1). We further use the approach from Nieves et al., NatMeth, 20, pages 259–267 to simulate structures with blinking, see comment 1.3.

Changes in the manuscript 1.1

To fully address the comment of the reviewer in the revised version of the manuscript we have added a new section within “**Accurate extraction of individual assemblies across diverse biologically inspired growth types.**”:

“To further evaluate the performance of SEMORE on segmentation and analysis of dynamic SMLM data we performed a series of stress tests on simulated data. We firstly evaluated SEMORE's ability to track morphological changes in time using simulated data of 3 types of spatially overlapping protein clustering morphologies with temporal information included. Snapshots of SEMORE's clustering in time provide visual confirmation of SEMORE's ability to track morphological changes in time (see Supplementary Fig. 5 for snapshots of simulated and Supplementary Fig. 6 for experimental data). Further quantification of SEMORE's

ability to accurately track and segment spatially overlapping protein assemblies in time reveals growth onset times across 3 morphological classes are predicted with an average offset of just ~13 frames (Supplementary Fig. 6). Subsequently, we test the clustering module on simulated, sparse structures containing as little as 4, 8, 15 and 25 point detections akin to data of protein oligomerization (see Supplementary Fig. 7 & 8). SEMORE analyzes the entire field of view at once and extracts structures down to 4 detections while maintaining >90% accuracy at biologically relevant noise to signal ratios. The morphological fingerprinting module can further refine the false positive detections from noise with as little as 4 detections, achieve full separation from noise at 8 detections and classify morphological classes in true structures at 15 detections (see Supplementary Fig. 8 & 9). Lastly we evaluated its performance to degenerative structures that shrink in time, akin to protein depolymerisation. SEMORE accurately segments 3 anisotropically degenerative morphologies showcasing it can be used to analyze dynamic shrinkage or depolymerisation of protein clusters (see Supplementary Fig. 10). In essence SEMORE only requires <10 detections to accurately segment and classify heterogeneous structures with dynamic morphologies further demonstrating its operational utility and potential for 4D SMLM.”

Elaborated in **results** on the scale invariance:

“The clustering module of SEMORE consists of multiple steps that self-parameterize based on the input data and are scale invariant due to the inherent 3D axis standardization (see Methods). This is designed to account for the inherently heterogeneous nature of biological assembly systems, in size, scale, spatial overlap, density, and morphology as well as the variability across experimental configurations that challenges current analytical tools.”

In addition, to address the comment we have added 5 new SI figures, namely Response figures: 1,4,5, 6, and 7.

Comment 1.2

- It would be interesting to know a little more about the limits of SEMORE, as in biological SMLM data there may be several different classes of cluster, which may have stronger similarities, within the same ROI. For example, if a protein is present as a monomer and can assemble into higher order oligomers, it might be expected there are some low-level oligomers (4 proteins clustered proteins at sub-diffraction, e.g., tetramers) in the data at time of fixation. Would these small clusters be distinguishable from the noise class? Some examples of how SEMORE deals with this low-level oligomerisation (low-density clusters) would strengthen the manuscript.

Response 1.2

This is indeed a valid comment and we thank the reviewer(s) for allowing us to address this in detail. As we detailed in response to your comment 1.1 in the revised manuscript SI fig 7 (response figure 4) we detail how SEMORE performs for low-level oligomers at sub-diffraction limit. The data demonstrate that SEMORE captures early stages of protein cluster initiation, even at diverse noise conditions. We also demonstrated how it dissects heterogeneous morphologies.

Changes in the manuscript 1.2

Please see changes as per comment 1.1 as we grouped our response and changes to this comment. In short, a new section on SEMORE’s performance on small sparse clusters showcases the potential use of SEMORE on tetramers or early stages of clustering.

Comment 1.3

- Following on from this point, one thing lacking from the manuscript is any comment on a common problem for arguably the two most common SMLM techniques (PALM, and dSTORM - used here); multiple blinking of the fluorophore. Normally, this introduces artefactual clustering into the SMLM data, and several approaches now exist for correcting this (e.g., Jensen et al., NatMeth, 19, pages 594–602, 2022, and Bohrer et al., NatMeth, 18, pages 669–677, 2021). Thus, in relation to biological data, some structures that may appear clustered, and then segmented in uncorrected data, will not be so after correction, shifting the observed distributions of the classes. Would the authors recommend such corrections before use of SEMORE for accurate fingerprinting? Also, some sense of how the lack of blink correction could impact classification would be beneficial. This could be done by comparing a ground truth of the molecule positions and a version where the data has added blinking.

Response 1.3

This comment on blinking is indeed very valuable. To answer and quantify SEMORE on data with and without blinking we rely on the approach Nieves et al., NatMeth, 20, pages 259–267. SI figure (Supplementary Fig. 9, response Fig. 10) shows SEMORE's performance on data with and without simulated blinking. The results show SEMORE achieves extraction accuracies above 90% and produces morphological fingerprints that reliably differentiates morphological classes from just 15 detections regardless of blinking. While we would recommend end-users to correct for blinking prior to using SEMORE, SEMORE performs well regardless of blinking and simply requires a small number of detections as seen in the response and in response 1.1+1.2.

Changes in the manuscript 1.3

The revised manuscript now includes a SI figure (**Supplementary Fig. 9, response Fig. 10**) showing performance with blinking and a discussion on blinking and our recommendation. We highlight the small effect of blinking on morphological fingerprinting in the section: **“Morphological fingerprinting captures defining features separating heterogeneous assemblies.”** :

“Blinking is a common challenge in SMLM, especially for PALM and dSTORM, and may lead to artificial clustering² and misinterpretation of protein assembly morphology. We map the effect of blinking on the morphological fingerprinting module using data with and without simulated blinking (see Methods). SEMORE remains largely unaffected by blinking achieving segmentation accuracies above 90% and reliably differentiates morphological classes from just 15 detections regardless of blinking (see supplementary Fig. 9). In addition to blinking, protein assemblies can exhibit biological behavior involving significant morphological changes (e.g., spherical to tubular), or as discussed in the evaluation of the clustering module above, depolymerization. To this end, we evaluated SEMORE's ability to track the temporal evolution of morphology between diverse morphologies, i.e. from fibril-like to spherical to asymmetric. SEMORE clearly tracks morphologies in time and transitions while outputting the most distinguishing features of each morphology offering important mechanistic insights (supplementary Fig 15).”

In addition, we include in the **discussion** our recommendation for blinking or other aberration:

“SEMORE is currently optimized for laterally stable assemblies, albeit the temporal refinement does handle lateral movement below the agnostic, data-derived search range defined per assembly basis (see Methods). Future versions are planned to include our diffusional analysis framework for motion aware clustering analysis. Blinking has insignificant effect on SEMORE analysis, however we recommend common aberrations such as

blinking and vibrational or spherical aberrations to be corrected prior to use of SEMORE to avoid any potential misinterpretation of the data.”

Minor comments:

- All figures showing SMLM data or plots of clusters etc. need a scale bar or some axes. It was hard to appreciate, until using the code, how different the length scale of the clusters between simulations and experimental data were (some several microns, whilst NPC data is much smaller scale)

Thank you for noticing the missing element. We note that as we highlighted in response to comment 1.1 SEMORE is purposefully designed to be dimension independent therefore the simulated data are without a scale bar. We do, based on your comment 1.1 elaborate on the scale-invariance in the main text. Furthermore figures with experimental data have been rectified in the revised version to include a scale bar as seen in main figure 4 making the size difference between insulin aggregates and NPC easier to appreciate.

- I ran the SEMORE installation as directed on the GitHub, generating a new environment. I found quickly that I was missing the dependencies needed to run the scripts (e.g., opencv in the dependency file should be opencv-python, I assume this is why this file may not have worked for me). Further, other more basic packages are of course not there in a new environment, e.g., pandas, numpy, etc., and require the user to add them. I would recommend the authors give a full and correct dependencies file, as this troubleshooting would be beyond many of the potential users with limited coding experience.

We have now provided the packages as a dependency.yml to be installed with:

“

```
git clone https://github.com/hatzakislabs/SEMORE
cd SEMORE
conda env create -f dependency.yml
conda activate SEMORE
```

”

and extended the documentation on the GitHub page. We appreciate the test of the github structure which now should be in order but we also include contact information and are happy to assist future users.

Additional minor changes

minor spelling and phrasing mistakes were rectified.

minor addition to account for the fact that we have 5 experimental data sets instead of 2 in the original submission, as well 6 additional ground truth simulated data sets.

addition of a few extra refs to account for reviews comments.

Reviewer 2

Bender et al. present SEMORE, SEgmentation and MORphological fingErprinting, as a semi-automatic machine learning framework for the analysis of super-resolution data to extract and quantify protein structures. By implementing a multi-layered density-based clustering module and a morphology fingerprinting module for quantification, they demonstrated SEMORE on simulations and experimental super-resolution data, including

insulin aggregates and nuclear pore complexes. Developing automatic clustering analysis without a priori knowledge and tracking the temporal evolution of morphology changes of protein clustering from single-molecule localization microscopy image data is certainly a good direction for the development of super-resolution microscopy field. **However, in my view, their demonstrations are limited to the analysis of the growth of immobile clusters under well-controlled noise conditions, which could be quite different from general biological behaviors.** This raises concerns about the limited applicability of this method. Additionally, although the authors insist that their method represents the first time-aware framework for 4D super-resolution data, **they mainly focus on presenting the classification results for protein cluster growth, without showing identified cluster images at each time point.** Therefore, it is hard to judge whether their method can effectively capture the temporal changes of protein clusters. To claim that their method is genuinely powerful for automatic clustering analysis and tracking the temporal evolution of protein clusters, they should also show the time-resolved clustering data. Furthermore, I cannot see **the significant novelty of this method; they mainly used the reported clustering method (DBSCAN) for cluster identification.** Although it appears that they additionally conducted temporal refinement after the DBSCAN analysis, this additional step seems to be mainly for dissecting clusters over time, rather than advancing the clustering analysis method. The new development of their method seems to be a morphology fingerprinting module that uses multiple feature vectors for quantification. **However, the performance of this method for real experimental time-resolved super-resolution image data was demonstrated only for the relatively new imaging method REPLOM, which is not a general type of data.** To demonstrate the general performance of this method, the authors should present more general time-resolved single-molecule localization microscopy data, rather than static dSTORM data. Although the authors applied their method to NPC data, it appears to be just static data, and the categorization of the clusters is not biologically meaningful (individual, overlapped, noise). For these reasons, I believe the authors' claims are not well supported, which unfortunately does not warrant publication in Nature Communications in the current form of the manuscript.

General response

We thank the reviewer for critically reading the manuscript, acknowledging that the approach we are using is certainly a good direction for the further development of the super-resolution microscopy field and providing valuable feedback. We have addressed all your comments explicitly, and in full detail by:

- a) Demonstrating SEMORE's performance on two additional experimental time-resolved super-resolution data based on sptPALM and live-cell PALM. Results show in both cases that SEMORE successfully clusters detections, captures temporal morphology changes, quantification of clusters by fingerprinting offering potential new insights. SEMORE is validated by comparison to results of the original work on these data sets; Wallis et al ³ and Hou et al⁴.
- b) Demonstrated SEMORE on additional experimental dSTORM data containing consecutive frames. Results show SEMORE obtaining comparable results in

clustering as Nieves et al¹ while temporal refinement and smart density filter to enable the additional pruning of false positives.

- c) Performing additional multiple noise variation: Simulating additional heterogeneous noise conditions with varying densities and displaying the capacity of SEMORE to accurately segment and classify them. Thus, the revised version now contains 2 independent experimental data sources with biological noise along with simulations containing homogeneous and heterogeneous noise. New noise types was also supported by additional experimental data as per a and b
- d) Displaying snapshots of SEMORE's segmentation at multiple time points during the assembly process for experimental data and simulated data. Followed by an evaluation of the number of detections needed to accurately detect the initiation point of protein assemblies.
- e) Displaying SEMORE's ability to accurately capture morphological variations as they evolve in time. Followed by an evaluation of the number of detections required to classify morphological classes.
- f) Detailing how SEMORE significantly advances the field: Explained in detail the multiple components incorporated in SEMORE prior and on top of DBSCAN, making DBSCAN a small part of one of the two modules of SEMORE. Compared SEMORE to the state of the art (DBSCAN) and detailed how the clustering module (segmentation) along with the fingerprinting module (quantification and classification) of SEMORE can propel the field of super-resolution. We stress here that we already had in the original submission a comparison of SEMORE to DBSCAN clearly demonstrating the superiority of SEMORE along with the increased accuracy following our developed smart noise filter.

However, we respectfully disagree that REPLOM "... is not a general type of data" REPLOM represents one of SMLM approaches enabling temporally resolved observation of protein aggregation. While indeed the implementation is novel and different to more common approaches the actual data type is very similar , if not identical. REPLOM is based on TIRF microscopy to observe isolated PSFs to accurately fit subpixel localizations in essence outputting a set of xy and t, coordinates.

The wealth of extra analysis and additional data, which fully and further support our claims, helped us improve the technical quality of the manuscript, benchmark it against state of art, and elaborate on the novelty and applicability of the method to extract mechanistic insights across a spectrum of diverse systems. We do hope the following point-by-point response and revised manuscript rectify all comments and convince the reviewer to join the other 2 reviewers in accepting the manuscript for publication. Figures created following your and other reviewers comments have been pasted below for your convenience.

Comment 2.1

1. As suggested above, the main problem with the current manuscript is the lack of time-resolved analyzed data. They only showed the classification results from simulation data for isotropic, random, and fibril growth. However, the **time-resolved cluster morphological change data (at each time point) should also be presented to assess whether their method correctly identified each growth pattern**. Additionally, I would like to see the time-scaled morphological changes of insulin aggregates and nuclear pore complexes from experimental data to evaluate whether their method performs well even with real data under heterogeneous noise conditions.

Response 2.1

This is a great set of questions and we thank you for giving us the chance to fully and in detail address them. In the revised version we now provide:

- a) 3 additional experimental data of 1) dynamic dSTORM 2) sptPALM and 3) live-cell PALM data sets and multiple additional simulations producing 11 additional SI figures. In all cases SEMORE precisely analyses and classifies the dynamic data sets showcasing it can be used as a universal tool (Supplementary Fig. 17 & 18 & 19, response figures 1 & 2 & 3).
- b) Snapshots of the experimental data of temporally resolved insulin aggregation displaying the temporarily resolved morphology readouts and the classification at different time points of the clustering by SEMORE and display SEMORE's prediction of the cluster initiation points (Supplementary Fig. 6, response Fig. 5). We stress that while in the original submission we displayed only one of the frames of the temporally resolved simulations or experimental data sets, the temporal refinement was existing and a key element of SEMORE and therefore underlie all figures. We thank the reviewer for noticing the insufficient explanation
- c) Additional simulation of temporarily resolved clustering data containing 3 types of growth morphologies. This type of simulation was already included in the original submission but not adequately discussed in the context of morphology detection early in the clustering process and in time. The data in the revised Supplementary Fig. 5, (response Fig. 7) show how SEMORE captures the temporal evolution of the 3 morphologies in time.
- d) Simulated dynamic morphology variation of a cluster in time by having a cluster sequentially express diverse morphologies, Supplementary Fig. 15, (response Fig. 8). The protein cluster is shown to drastically change morphology from fibril-like structures to asymmetric structures and also to spherical structures, each structure consisting of 200 data points. The temporal morphology changes were analyzed by the fingerprinting module of SEMORE and the extracted features embedded in the 2-component map resulted in a clear separation of the diverse morphologies. Thus, the new Supplementary Fig. 15, (response Fig. 8) shows how SEMORE accurately captures diverse morphologies in time. To further demonstrate the potential of SEMORE we subjected the fingerprints to a MinMax normalization and plotted them in a heatmap. As expected the morphological fingerprint not only accurately classifies the diverse temporally separated morphological growths, but outputs the geometric features that drive this classification offering further mechanistic insights. In addition,

response 1.1+1.2 shows only 15 detections are needed for this accurate classification of morphology.

- e) Single frame of the entire clustering process for 3 diverse morphologies occurring in parallel and in a single field of view Supplementary Fig. 6, (response Fig. 6). SEMORE predicts the growth onset times with an average offset of just ~13 frames thus accurately predicting the clustering initiation point for all three morphology classes despite them being highly heterogeneous and spatially overlapping. Panel d in new SI Supplementary Fig. 6, (response Fig. 6) displays the time from growth start to SEMORE initiation prediction highlighting that not only SEMORE predicts cluster morphology but also the initiation point accurately.

The reviewer also asked for time-scaled morphological changes of nuclear pore complexes, albeit such data may have been produced we do not know of its existence and have not been able to locate it. The NPC data serves to demonstrate the use of SEMORE on static data and in an independent experimental setting. We addressed the comment on the classification of time-resolved morphological variations on experimental and simulated data by the a-c) above.

To address the comment of the reviewer on the heterogeneous noise we provide new simulation on diverse noise conditions. In the original manuscript SI fig. 2, (still SI fig 2) we had a noise stress test by varying the noise-to-event ratio by more than an order of magnitude (45 fold). We now extended the stress test to apply, and account for, non-uniformly distributed noise. We simulated an additional 5-25 noise seeds each containing 20-50 points and also applied Gaussian distributed shifts to all initial noise points further reducing the uniformity (See new Supplementary Fig. 3, (response Fig. 9)). We are happy to report that the classification accuracy of SEMORE remains practically unaffected and more than 85% for all biological relevant noise levels.

The response to 1.1 and 1.2 further emphasizes the ability of SEMORE to capture small sparse structures which extends to show how SEMORE performs in early stages of oligomerization.

Changes in the manuscript 2.1

To fully address the comment of the reviewer we:

1. Introduced 7 new Supplementary figures that fully and detail address the comment of the reviewer but also include comments of the other two reviewers.
2. Added three new SI figures (SI fig. 17+18+19, response figures 1+2+3) showcasing SEMORE on additional experimental data along with comments in the revised version of the manuscript:

“We further demonstrated SEMORE’s ability to utilize the temporal dimension in experimental data using three additional published data sets: Firstly, dSTORM data of fibroblast growth receptor ¹ ¹, where SEMORE utilizes the data’s reported frames for temporal refinement together with smart density filtering to accurately extract protein clusters, prune false positive detections and provide cluster size estimation (Supplementary Fig. 17). Secondly, super-resolved dynamic data syntaxin 1a spatial clustering by sptPALM ³, where SEMORE accurately captures clusters, their morphological evolution in time (Supplementary Fig. 18). Thirdly, super-resolved live-cell PALM data of ryanodine receptors (RyRs) ⁴. SEMORE accurately captures RyR clusters with a granularity infeasible by current methods such as DBSCAN alone (Supplementary Fig. 19). The fact that SEMORE operates across dynamic experimental

SMLM data outputting clusters and their properties in agreement with published analysis, further highlights its potential as a robust universal tool for 4D cellular biology. ”

3. Inserted and elaborated the section “**Precise extraction and quantification of experimental super-resolution data.**” in the main text describing SEMORE’s ability to perform accurate extraction of protein assemblies in time for experimental data:

“Studies of aggregating insulin using the super-resolution approach REPLOM, offering real-time recordings of the aggregation growth, revealed insulin aggregates by two distinct pathways: an isotropic pathway, generating radially grown spherical structures and an anisotropic pathway, generating more elongated asymmetric structures consistent with the presence of multiple nucleation sites. When the raw x, y, t coordinates of insulin aggregation events were fed to SEMORE, the clustering module automatically extracted 139 individual insulin aggregate structures (Fig. 4a) despite them displaying large heterogeneity in their morphology. The identified structures are largely spatially overlapping, an element that would challenge their proper separation by manual inspection or current tools. However, the implementation of the temporal element of SEMORE capturing time-dependent morphological evolution allowed their rapid and precise spatial separation, thus mitigating potential morphology misinterpretation (Fig. 4a and Sup Fig 5). Snapshots of the successful capture of insulin aggregation during its growth combined with the insulin aggregation initiation time point identification by SEMORE are displayed in supplementary fig. 6.”

In addition, to further probe SEMORE’s ability to utilize the temporal dimension in experimental data SEMORE was evaluated on three additional data sets: Firstly, using dSTORM data of fibroblast growth receptor 1¹ SEMORE utilizes the data’s reported frames for temporal refinement together with smart density filtering to accurately extract protein clusters, prune false positive detections and provide cluster size estimation (Supplementary Fig. 17). Secondly, utilizing super-resolved dynamic data syntaxin 1a spatial clustering by sptPALM³ SEMORE accurately captures clusters, their morphological evolution in time and provides potential for new insights on clusters by morphological fingerprinting (Supplementary Fig. 18). Thirdly, super-resolved live-cell PALM data of ryanodine receptors (RyRs) ⁴ SEMORE accurately captures RyR clusters with a granularity infeasible by current methods such as DBSCAN alone. Lastly, SEMORE operates across dynamic experimental SMLM further highlighting its potential as a robust universal tool for 4D cellular biology.

4. Inserted a section in the main text in “**Accurate extraction of individual assemblies across diverse biologically inspired growth types.**” describing SEMORE’s ability to perform accurate extraction of protein assemblies in time for simulated data consisting of spatially overlapping structures across 3 morphology classes:

“To further evaluate the performance of SEMORE on segmentation and analysis of dynamic SMLM data we performed a series of stress tests on simulated data. We firstly evaluated SEMORE’s ability to track morphological changes in time using simulated data of 3 types of spatially overlapping protein clustering morphologies with temporal information included. Snapshots of SEMORE’s clustering in time provide visual confirmation of SEMORE’s ability to track morphological changes in time (see Supplementary Fig. 5 for snapshots of simulated and Supplementary Fig. 6 for experimental data). Further quantification of SEMORE’s ability to accurately track and segment spatially overlapping protein assemblies in time reveals growth onset times across 3 morphological classes are predicted with an average offset of just ~13 frames (Supplementary Fig. 6).”

5. Inserted a section in the main text within “**Morphological fingerprinting captures defining features separating heterogeneous assemblies.**” describing SEMORE’s ability to extract time-resolved morphology information for both simulated and experimental data: *“In addition to blinking, protein assemblies can exhibit biological behavior involving significant morphological changes (e.g., spherical to tubular), or as discussed in the evaluation of the clustering module above, depolymerization. To this end, we evaluated SEMORE’s ability to track*

the temporal evolution of morphology between diverse morphologies, i.e. from fibril-like to spherical to asymmetric. SEMORE clearly tracks morphologies in time and transitions while outputting the most distinguishing features of each morphology offering important mechanistic insights (supplementary Fig 15)."

6. Inserted a section in the main text "**Accurate extraction of individual assemblies across diverse biologically inspired growth types.**" on SEMORE's ability to extract protein assemblies consisting of few detections to evaluate SEMORE's ability to capture small assemblies and early stages of assemblies.
"Subsequently, we test the clustering module on simulated, sparse structures containing as little as 4, 8, 15 and 25 point detections akin to data of protein oligomerization (see Supplementary Fig. 7 & 8). SEMORE analyzes the entire field of view at once and extracts structures down to 4 detections while maintaining >90% accuracy at biologically relevant noise to signal ratios. The morphological fingerprinting module can further refine the false positive detections from noise with as little as 4 detections, achieve full separation from noise at 8 detections and classify morphological classes in true structures at 15 detections (see Supplementary Fig. 8 & 9)."
7. We modified an existing section "**Accurate extraction of individual assemblies across diverse biologically inspired growth types.**" discussing noise to the following
"The median classification accuracy of SEMORE remained above 85% for a range of noise density levels and was practically independent of noise being homogeneous or heterogeneous (see Methods, Supplementary Fig. 2 + 3)."
8. We modified an existing section "**Precise extraction and quantification of experimental super-resolution data.**" discussing noise to the following
*"The summary of these results on a diverse set of experimental systems demonstrates the ability of SEMORE to generalize to completely different sets of biological systems, imaging and experimental conditions, noise types, and molecular scales without any *a priori* knowledge of their structure."*
9. Added in SI fig. 3 of the original submission now SI fig. 4 that data represents a single frame of a temporal simulation.
10. Highlighted in main figures 1+3+4 in revised submission when shown data represent a single frame of a temporally resolved experiment.
Main fig. 1 added: *"... temporally resolved insulin aggregation imaged using the REPLOM⁵ approach on a TIRF microscope..."*
Main fig. 3 added: *"Three classes of time-resolved aggregations were simulated to capture a broad aspect of biological systems (see Methods)"*
Main fig. 4 added: *"Top: Final frame of accumulated super-resolution localizations from temporally resolved insulin aggregation"*
11. In Methods we have elaborated and extended the relevant sections to include these new performance evaluations.

Comment 2.2

The demonstrated data seems to only focus on the growth of clustering. However, changes in protein clusters in the real world could involve not only cluster growth but also significant morphological changes (e.g., spherical to tubular), shrinkage, movement, and scattering (depolymerization). For wider applicability of this method, the authors should include related data as well.

Response 2.2

We indeed fully agree with the reviewer that time-dependent morphological variations are an important element to study. As a matter of fact capturing temporal morphological variations

are inherent in SEMORE and underlie accurate extracting spatially overlapping structures as highlighted in response 2.1. However, we agree this could not be fully appreciated in the original manuscript.

We thank the reviewer for allowing us to rectify this, expanding the showcasing of SEMORE's capabilities.

- For answering SEMORE's performance of dynamic morphological changes please see answer to this reviewer comment 2.1c, and Supplementary Fig. 15, (response Fig. 8).
- Regarding the reviewer comment on lateral movement: SEMORE considers as input the set of $XY(z)$, and t , coordinates that common SMLM software outputs. As such currently, SEMORE does not include the tracking of large lateral movement beyond what can be captured by temporal refinement, i.e. large lateral shifts (larger than the cluster dependent radius calculated per cluster, see Methods) between frames would currently be recognized as a new cluster. This is now explained in the discussion of the revised version.
- Protein depolymerisation and cluster shrinkage is indeed an important element underlying biological processes and we are thankful to the reviewer for bringing this up. To evaluate how SEMORE accounts for shrinkage and depolymerisation of protein clusters we reversed the simulation of growth of diverse morphologies. In the revised Supplementary Fig. 10, (response Fig. 11) we display a simulated data set of depolymerisation as well as the clustering of these shrinking structures as they are predicted by SEMORE. The accurate clustering further supports the strength of SEMORE to also be capable of clustering shrinking protein clusters.

Changes in the manuscript 2.2

Following the reviewers comment we have strengthen the manuscript with the following:

- 1) Protein depolymerisation and shrinkage: In the section "**Accurate extraction of individual assemblies across diverse biologically inspired growth types.**" discussing the dynamic nature of SEMORE we added: "*Lastly we evaluated its performance on degenerative structures that shrink in time, akin to protein de-polymerisation. SEMORE accurately segments 3 anisotropically degenerative morphologies showcasing it can be used to analyze dynamic shrinkage or depolymerisation of protein clusters (see Supplementary Fig. 10).*".
- 2) Lateral displacement: We added in the **discussion** "SEMORE is currently optimized for laterally stable assemblies, albeit the temporal refinement does handle lateral movement below the agnostic, data-derived search range defined per assembly basis (see Methods). Future versions are planned to include our diffusional analysis framework for motion aware clustering analysis. Blinking has insignificant effect on SEMORE analysis, however we recommend common aberrations such as blinking and vibrational or spherical aberrations to be corrected prior to use of SEMORE to avoid any potential misinterpretation of the data."
- 3) Added a new Supplementary Fig. 10, (response Fig. 11) displaying how SEMORE accurate clusters protein assemblies undergoing depolymerisation for 3 diverse structures.
- 4) As reported in changes in the manuscript for comment 2.1 we have added in the revised version a section on SEMORE's ability to capture temporal variations in morphology,

Comment 2.3

Their demonstration is mostly based on simulation data, where the noise appears to be well-controlled and homogeneous. Although they tested this method under different noise density conditions, it seems that the density they used is uniformly distributed across the entire field of view and throughout the entire time domain, which could differ from real conditions. Additionally, the intensity of noise could be heterogeneous. Because such heterogeneity of noise could affect the identification and classification of protein clusters, they should test the method with various types of noise.

Response 2.3

The reviewer is correct and we are thankful for bringing this to our attention. As we outlined in the response to their comment 2.1 we have now extended the stress test to apply, and account for, noise that is non-uniformly distributed in space. We simulated an additional 5-25 noise seeds each containing 20-50 points and also applied Gaussian distributed shifts to all initial noise points further reducing their uniformity (See new SI fig. 3, response fig. 9). We are happy to report that the classification accuracy of SEMORE remains practically unaffected and more than 85% for all biological relevant noise levels. We have also analyzed 3 additional experimental sets and we are happily reporting SEMORE accurately handles real experimental data sets with diverse and heterogeneous noise levels (see detailed answer to comment 2.1)

Changes in the manuscript 2.3

- 1) In the main text in the section discussing the noise simulation we added:
“*The median classification accuracy of SEMORE remained above 85% for a range of noise density levels and was practically independent of noise being homogeneous or heterogeneous (see Methods, Supplementary Fig. 2 + 3).*”. In support we added a new supplementary figure 3 displaying the benchmarking for non-uniform noise addition.
- 2) Added analysis and results for experimental data see detailed answer to comment 2.1.

Comment 2.4

The authors claim that the use of a machine learning-based method is an important aspect of this study. However, the machine learning method employed in this paper appears to be limited to unsupervised point clustering, rather than classification or other methods. Moreover, these unsupervised point clustering methods used here do not seem to differ significantly in terms of methodology from the existing method, DBSCAN. I believe the authors have not sufficiently explained why the method they used can be considered a machine learning approach and what sets it apart as a novel contribution compared to existing methods

Response 2.4

We respectfully, but firmly, disagree with the reviewer that the method does not differ from a DBSCAN. In fact, DBSCAN is only a small part of the multimodular part of SEMORE and we indeed had compared the operational performance of SEMORE to DBSCAN in SI fig. 3 of the original submission (now SI fig. 4). Indeed, SEMORE achieved a 89% median accuracy as compared to 70% for DBSCAN alone on simulated data, convincingly demonstrating the

superior performance of our multimodular pipeline. For experimental data SEMORE is key to accurate analysis as DBSCAN alone can not provide the granularity required to dissect small clusters of ryanodine receptors (See new SI fig. 19, response fig. 3). An additional key example is extracting biologically correct structures and features from insulin aggregation which could otherwise lead to misinterpretation (see main figure 1 for comparison of before and after temporal refinement or SI fig. 3 of the original manuscript (now SI fig. 4). Combined these data showcase the entirety of the SEMORE clustering pipeline is required to extract spatially overlapping structures, provide granularity, filter out noisy detections as clusters that may in the case of no assemblies in the field of view produce nonsensical structures.

Several key features sets SEMORE beyond DBSCAN and current state of that art ::

- a) Automatic, data-driven model selection between HDBSCAN and DBSCAN based on the density of localisations computed using the field of view.
- b) Semi-supervised hyperparameter choices based on experimental data.
- c) Topological failsafe for the case of no initial aggregation securing the clustering module does not form nonsensical clusters as DBSCAN alone might if only presented with noise, thus enhancing the clustering detection sensitivity.
- d) Smart noise filtering filtering false positive detections in conjunction with our clustering greatly improves the quality of the clustering module's output.
- e) Temporal refinement enables clustering of spatially overlapping structures, the dissection of initiation point of each cluster in time and following of morphology changes over time all of which are currently infeasible by DBSCAN or other available methods alone.
- f) Morphological fingerprinting for quantification and ranking of interpretable geometric and kinetic descriptors, that to the best of our knowledge is introduced here by us for protein clustering, based on our earlier work on diffusional fingerprinting.

This combination extends above and beyond any current methods and allows SEMORE to yield automatic segmentation, classification and quantitative insights on arbitrarily complicated data sets across experimental conditions of protein clustering by SMLM and the temporal evolution of morphology without a priori knowledge.

We respectfully emphasize that a machine learning pipeline based solely on unsupervised learning is still a machine learning approach and an approach that can generalize to problems outside the training distribution. Indeed, unsupervised machine learning methodologies are routinely used in recent interdisciplinary publications^{6 7 8}. Moreover, we are slightly puzzled by the comment that no classification or other machine learning approaches are present in the original manuscript: We use a) an additional DBSCAN in the UMAP embedded space (unsupervised) to classify structures in original main fig. 3 and original SI fig. 4-6 (now SI figs. 11-13), b) a boosted decision tree (supervised) in SI fig. 6 (now new SI fig. 13) to highlight the potential use of morphological fingerprints in a downstream model, c) a Linear Discriminant analysis (supervised) in feature importance

ranking SI. fig. 8 (now SI fig. 16). To emphasize these important elements in the revised manuscript we elaborate on the use of the unsupervised output in supervised settings.

Changes in the manuscript 2.4

To fully address the comment of the reviewer we added a complete description of the elements that set SEMORE above and beyond the current state of the art. In the revised version, we have added:

- 1) an extension to the explanation of SEMORE's clustering module in the **results**: *"The clustering module of SEMORE consists of multiple steps that self-parameterize based on the input data and are scale invariant due to the inherent 3D axis standardization (see Methods). This is designed to account for the inherently heterogeneous nature of biological assembly systems, in size, scale, spatial overlap, density, and morphology as well as the variability across experimental configurations that challenges current analytical tools. The pipeline initially inspects high-density areas in a standardized Euclidean 3D space, using a hyperparameter space pre-defined for this region, and provides an appropriate model choice based on a data-driven decision. The chosen density-based scanning model, either HDBSCAN or DBSCAN, extracts high-density regions of biomolecules (clusters or aggregates) from low-density regions (noise) (Fig. 1b). The initial clustering contains an added topological fail safe to prevent detection of nonsensical structures (see Methods). If a temporal dimension is available, the high-density regions are treated through our temporal refinement (Fig. 1c). Segmentation in time and temporal refinement is strictly required to dissect spatially overlapping structures within high-density areas. The clustered output is further refined by subjecting all identified assemblies to a smart density filter to eliminate falsely predicted assemblies that do not meet an agnostic, data-derived density criteria (Supplementary Fig. 1). The result is a robust clustering model outcompeting current methods and building towards the first general-purpose approach for dynamic SMLM (see Methods and Supplementary fig. 4)."*
- 2) The methods section of the revised manuscript have been elaborated to further highlight the key contributions of the SEMORE clustering pipeline mentioned in a-f of the response.
- 3) We highlight the gain of SEMORE in accurate segmentation of spatially overlapping structures as compared to current methods on experimental data in section **"Precise extraction and quantification of experimental super-resolution data."**: *"When the raw x, y, t coordinates of insulin aggregation events were fed to SEMORE, the clustering module automatically extracted 139 individual insulin aggregate structures (Fig. 4a) despite them displaying large heterogeneity in their morphology. The identified structures are largely spatially overlapping, an element that would challenge their proper separation by manual inspection or current tools. However, the implementation of the temporal element of SEMORE capturing time-dependent morphological evolution allowed their rapid and precise spatial separation, thus mitigating potential morphology misinterpretation (Fig. 4a and Sup Fig 5)."*
- 4) Elaborate on the use of the unsupervised output of SEMORE in a supervised setting within **"Morphological fingerprinting captures defining features separating heterogeneous assemblies."**: *"We evaluated how morphological fingerprinting can dissect the diverse types of otherwise similar morphologies by three approaches. Firstly, utilizing a second UMAP embedding and DBSCAN of the circularity feature subset offered additional investigation of the fibril cluster in the embedded fingerprint space (see fig. 3b) . This resulted in two spatially separated clusters corresponding to branching and non-branching fibrils which independently can be achieved by boosted decision tree classification using all fingerprint features directly (see Supplementary Fig. 13). Secondly, by investigating the isotropic and anisotropic clusters which revealed more continuous spaces given their more smooth growth behaviors as compared to branching of fibrils, yet with clear spatial separation of fingerprint features (see Supplementary Fig. 14). Lastly, SEMORE was able to correctly classify the identity of diverse morphologies in high-density regions reaching an F1 score of >98% (See Supplementary Fig. 11). Although, morphological fingerprints represent unsupervised output these results demonstrates the versatility in supervised classification to extend beyond distinguishing between fundamentally different morphology families, i.e., fibrils vs isotropic, to also capture heterogeneity within the same morphology family i.e., branching fibrils vs non-branching fibrils. We find the morphological*

fingerprinting needs just 8 detections to fully separate true detections from noisy and 15 detections to further classify the morphology class of the true detections (Supplementary Fig. 7 & 8). Such expressive power is required to provide mechanistic insights for most biological assemblies as they often follow one assembly mechanism but still exhibit heterogeneity in their final morphologies and the mapping of which is currently an analytical challenge⁹.”

Comment 2.5

In the Fingerprinting module, it is mentioned that over 40 features were utilized to analyze the properties of each cluster using various methods. However, it appears that a comprehensive assessment of the strengths and weaknesses of these methods was not conducted. It is unclear why the given methods are expected to yield good features for cluster classification. Additionally, it would be beneficial to determine if all the provided methods are necessary based on the analysis results. For instance, Supplementary Figure 8 suggests that it might be possible to identify a better method instead of using all the feature vectors obtained from each feature. There is a possibility that some of them could even interfere with the analysis process, so it would be insightful to see further results in this regard.

Response 2.5

The reviewer asks for the basis of the selection of the 40+ descriptive features and whether all features are needed, an assessment of their performance and whether their number can interfere with the analysis.

The central scope and strength of SEMORE is the maximization of applicability and the agnostic classification across diverse systems. Based on these criteria we created an extensive set of features that relied on a combination of geometric elements designed to best capture as diverse as possible morphologies and experimental systems. Therefore the number and identity of features should not be predetermined based on the classification of one specific structure type, but should be as wide as possible so as to ensure the reliability of the agnostic classification. Similarly the strength and weakness of these features can not be predetermined, as they are system dependent and will indeed vary from dataset to dataset.

We fully agree therefore with the reviewer that often only a fraction of the 40+ features can be sufficient to classify each of a given cluster morphology. We had provided this in original SI fig. 8 (now SI fig. 16) and extensively studied in the manuscript both on simulated and experimental data as per original main fig. 3+4 and original SI figures 2-8 and 11+12. To provide further evidence we now include additional simulated and experimental data as per new SI fig. 17+18+19 that shows the importance values of the most dominant features in all classifications of the original manuscript and clearly displays that some features are more important in each case. Each panel displays 18 of the features with the more dominant effect as the rest had minimal or no contribution to the classification, in agreement with what the reviewer asks. This figure stresses that *each morphology requires a distinct and different set of descriptive features* to be accurately classified. For example, for insulin classification, longest shortest distance (L_s_D), number of bridges in longest possible one-way route through graph (L_l_step), average number of neighbors in graph (mean K) are the dominant features while for Nuclear Pore Complexes longest shortest path in graph (L_s_path), ratio of longest distances in graph (L_l_ratio) and effectiveness of the longest shortest distance

(L_s_effectiveness) are now the most dominant ones. This is further highlighted in the new SI fig 15 (response figure 7) towards capturing temporal evolution of morphology that clearly displays classification of diverse morphologies relies on different features sets.

We also agree that some of the features can have overlapping interpretations. This overlap does not affect the overall accuracy of the method as only the effects that contribute to the classification are taken into account, however, they can collectively contribute to a detailed description of the clustering at hand.

We also agree that the features can always be improved and as better features are employed in the future, our implementation of morphological fingerprinting may be further extended even by users to create a feature library.

We are puzzled by the comment on the assessment of the descriptor importance as this is exactly displayed in SI fig. 8. Acknowledging that these may not be clear we have explained it in detail here and in the main text.

Changes in the manuscript 2.5

We agree with the reviewer on the importance of these comments surrounding features and features importance thus merits further clarification so we have highlighted the following elements in the main text revised version.

- 1) In the **Results** discussing the descriptive feature selection we added “ The number and identity of features are constructed as diverse as possible so as to agnostically capture a diverse set of protein clustering morphologies and maximize applicability across biological systems without *a priori* knowledge.”
- 2) In the same section: “Note that some of the features can have overlapping interpretations. This overlap does not affect the overall method accuracy as only features that enhance the classification are taken into account, however, they can collectively contribute to a detailed description of the system at hand. If better features are identified in the future, they can be conveniently implemented into SEMORE further extending its potential.”
- 3) Elaborated on the point of the possibility to extend the fingerprinting module: “*We envision SEMORE’s application across diverse systems, the continuous extension of the fingerprinting module and the generation of libraries of protein assembly morphologies. Libraries of morphological features could aid mapping of assembly characteristics to their identity and function for advancement of biological understanding, statistical approaches and be the basis for novel supervised learning purposes (Supplementary Fig. 13).*”
- 4) In the **Discussion** discussing the descriptive feature selection we added:
“Using a broad set of features maximizes applicability across biological systems and feature ranking can reveal the features of greatest importance providing key mechanistic insights into the given system”

Comment 2.6

In general, the methodology is not thoroughly explained. For instance, even if the method applied in the existing DBSCAN is used as is, it would be helpful if the methodology were described in more detail. Additionally, the methodology for the classification of each point, which occurs after the fingerprinting module and embedding with UMAP, is not elaborated very well. Since the fingerprinting module appears to be the only distinctive novelty of this method, as mentioned above, it should be described in detail to provide a clearer understanding.

Response 2.6

We appreciate the help to improve the readability of the methods section and for enabling us to improve the description of the element that sets us apart from the current state of the art. As outlined in our response to comment 2.4 we have in the revised version elaborated on the clustering module of SEMORE and its contribution to the field. The classification of fingerprints into specific morphology classes occurs by an additional DBSCAN on the 2-component UMAP embedded space (Main fig. 3 and original SI fig. 6c, now SI fig. 13c) or by a boosted decision tree (original SI fig. 6d+e, now SI fig. 13fd+e).

Changes in the manuscript 2.6

- 1) For the elaboration of the clustering module kindly see the changes of response 2.4. In addition
- 2) We have elaborated the methods section in the revised manuscript.
- 3) We have highlighted the use of DBSCAN and boosted decision trees in classification from morphological fingerprints: *“This resulted in two spatially separated clusters corresponding to branching and non-branching fibrils which independently can be achieved by boosted decision tree classification using all fingerprint features directly (see Supplementary Fig. 13). Secondly, by investigating the isotropic and anisotropic clusters which revealed more continuous spaces given their more smooth growth behaviors as compared to branching of fibrils, yet with clear spatial separation of fingerprint features (see Supplementary Fig. 14). Lastly, SEMORE was able to correctly classify the identity of diverse morphologies in high-density regions reaching an F1 score of >98% (See Supplementary Fig. 11). Although, morphological fingerprints represent unsupervised output these results demonstrates the versatility in supervised classification to extend beyond distinguishing between fundamentally different morphology families, i.e., fibrils vs isotropic, to also capture heterogeneity within the same morphology family i.e., branching fibrils vs non-branching fibrils.”*

Additional minor changes

minor spelling and phrasing mistakes were rectified.

minor addition to account for the fact that we have 5 experimental data sets instead of 2 in the original submission, as well 6 additional ground truth simulated data sets.

addition of a few extra refs to account for reviews comments.

Reviewer 3

The manuscript entitled “SEMORE: SEgmentation and MORphological fingErprinting by machine learning automates super-resolution data analysis” by Hatzakis et al introduces SEMORE, a semi-automatic machine learning framework for universal, system and input-dependent, analysis of super-resolution data. The manuscript is well written and informative. However, this manuscript can be improved by addressing the following issues:

General response

We thank the reviewer for critically reading the manuscript, acknowledging it is “informative” and “well written” and for proposing elements to further improve it. Fully and in detail addressing all of them helped us further improve the quality of the manuscript. Figures created following your and other reviewers comments have been pasted below for your convenience.

Comment 3.1

The authors have developed a new analytical tool SEMORE to extract and quantify underlying structures limiting single-molecule localization microscopy (SMLM). How this newly developed tool SEMORE is better suited with SMLM than some of the previously employed tools such as SMAP (modular super-resolution microscopy analysis).

Response 3.1

We are grateful for noticing the incomplete comparison of our method with the current state of the art. Super-resolution Microscopy Analysis Platforms for SMLM data (SMAP) is a platform carrying multiple tools for SMLM users to make SMLM more accessible. This modular analysis platform contains tools for localization, post-processing, rendering and clustering (specifically DBSCAN see below). The multiple steps involved in SEMORE's clustering module, i.e. the data-driven decision of clustering approach, smart noise filtering, temporal refinement and morphology fingerprinting do not have directly comparable approaches implemented in SMAP. SMAP does have a DBSCAN implemented, which we do provide a comparison to (original SI fig 3 now SI fig. 4). In addition, SMAP does not include a module comparable to SEMORE's morphological fingerprinting for the quantification of morphology. The methodological advances of SEMORE should be viewed as a possible addition to the multiple plugins already contained within SMAP. The potential of SEMORE to be added to SMAP or to work downstream of analysis performed in SMAP is now mentioned in the revised version of the manuscript to make future users aware of the synergy.

Changes in the manuscript 3.1

We thank the reviewer for enabling us to clarify the synergy between SEMORE and upstream analysis in SMAP. This has now been added to the revised version of the manuscript in the **discussion**, see

"In all cases, relevant structures are precisely extracted and featurized providing strong support for SEMORE as a universal, input-independent model as a convenient toolbox for the SMLM community to use in conjunction with or to be incorporated into SMAP."

For comparison of SEMORE to DBSCAN we refer to the original SI figure 3 (now SI fig. 4), where we show SEMORE achieves a 89% median accuracy as compared to 70% for DBSCAN alone.

Comment 3.2

The authors should explain how SEMORE deals with commonly occurring supervised and unsupervised machine learning problems.

Response 3.2

This is indeed an important element of our work, and we are grateful for allowing us to elaborate further. Below we have detailed the challenges of both supervised and unsupervised tools and how SEMORE addresses them.

Commonly occurring challenges in **unsupervised** machine learning may include:

- 1) Absence of direct accuracy evaluation because the data comes without any labels, classical performance evaluations such as accuracy is not possible.

- 2) The output of an unsupervised model may require human inspection to decipher how the model is performing.

Commonly occurring problems in **supervised** machine learning :

- A) Collection of labels for data is often a tedious and resource-strenuous process that requires extensive *a priori* expertise and is potentially subjected to unconscious biases.
- B) Supervised models are trained for a specific task and often require retraining when presented with a system outside the training distribution.
- C) Overfitting on data presents a large challenge to supervised approaches resulting in overly optimistic expectations on model generalization. Mitigation requires reliable validation and test schemes that are independent of the training distribution are not always feasible and require more manual labels to be curated.

SEMORE was specifically chosen to be an unsupervised approach so as to enable a universal approach for SMLM across experimental conditions and biological systems *without* the need for *a priori* knowledge or expensive label generation. We tackle the uncertainty surrounding the output of an unsupervised machine learning pipeline by extensive evaluations on simulated data with known labels and by probing performance at various perturbations. In addition, we evaluated its performance on 5 (2 in the original manuscript and now added 3 additional in new supplementary fig. 17-19) completely diverse experimental data acquired by different groups using diverse imaging and experimental conditions, on different biological systems and with varying noise. In all cases SEMORE's output is in agreement with the published outputs supporting it can surpass commonly occurring supervised and unsupervised machine learning problems.

Changes in the manuscript 3.2

To fully address the comments we discussed extensively in the revised version the a) actual challenge of supervised vs unsupervised analysis b) how we mitigated them. They are discussed in 3 areas of the manuscript

- a) Elaboration on common challenges in unsupervised and supervised learning see section: **Introduction**:

“Supervised algorithms are highly accurate when large amounts of annotated data are available albeit annotations require extensive manual labor and expert knowledge and the resulting model is often suitable for one specific data set or task. This imposes some challenges in exploring unmapped biological systems with no a priori knowledge and potentially limits their use as a general tool¹⁰⁻¹². Unsupervised approaches such as OPTICS¹³ and DBSCAN¹⁴ can overcome some of these limitations for coordinate-based input data. Their performance however is often limited by a one-size-fits-all approach. This often results in laborious human intervention in model tuning, restricting their adaptation to heterogeneity in localization densities and assembly sizes in varying experimental data^{15,16}.”

- b) To address how we mitigate the common challenges, see section **Introduction**:

“We show that SEMORE provides the unbiased unsupervised clustering, and morphological cluster variation in time, without a priori knowledge and for diverse simulated and experimental data sets: heterogeneous growth pathways of insulin aggregates, the dimensions of individual nuclear pore complexes, size of individual clusters of fibroblast growth receptors 1, temporal evolution of syntaxin 1a clusters and dynamic clustering of ryanodine receptors (RyR). The implementation of temporal dependence in morphological variations is a promising platform to handle static or dynamic super-resolution data and enables in-depth temporal-dependence analysis and segmentation of complex structures.”

- c) we highlight the use of supervised learning on the unsupervised output from SEMORE, see section: **Morphological fingerprinting captures defining features separating heterogeneous assemblies**: “We evaluated how morphological fingerprinting can

dissect the diverse types of otherwise similar morphologies by three approaches. Firstly, utilizing a second UMAP embedding and DBSCAN of the circularity feature subset offered additional investigation of the fibril cluster in the embedded fingerprint space (see fig. 3b) . This resulted in two spatially separated clusters corresponding to branching and non-branching fibrils which independently can be achieved by boosted decision tree classification using all fingerprint features directly (see Supplementary Fig. 13). Secondly, by investigating the isotropic and anisotropic clusters which revealed more continuous spaces given their more smooth growth behaviors as compared to branching of fibrils, yet with clear spatial separation of fingerprint features (see Supplementary Fig. 14). Lastly, SEMORE was able to correctly classify the identity of diverse morphologies in high-density regions reaching an F1 score of >98% (See Supplementary Fig. 11). Although, morphological fingerprints represent unsupervised output these results demonstrates the versatility in supervised classification to extend beyond distinguishing between fundamentally different morphology families, i.e., fibrils vs isotropic, to also capture heterogeneity within the same morphology family i.e., branching fibrils vs non-branching fibrils."

Comment 3.3

The authors should also highlight the role of SEMORE in the advancement of machine learning-based approaches.

Response 3.3

We value the comment of highlighting how SEMORE advances machine learning based approaches and how further elaboration can strengthen the manuscript. We highlight how a coordinate-based approach to the structural analysis coupled with the temporal refinement, the density-based modeling choice and data-based noise filtering enhances the machine learning pipeline. In addition, we emphasize how featurization of underlying structures directly from raw data can be used to build new machine learning frameworks by future users.

Several key features sets SEMORE beyond DBSCAN and current state of that art ::

- g) Automatic, data-driven model selection between HDBSCAN and DBSCAN based on the density of localisations computed using the field of view.
- h) Semi-supervised hyperparameter choices based on experimental data.
- i) Topological failsafe for the case of no initial aggregation securing the clustering module does not form nonsensical clusters as DBSCAN alone might if only presented with noise, thus enhancing the clustering detection sensitivity.
- j) Smart noise filtering filtering false positive detections in conjunction with our clustering greatly improves the quality of the clustering module's output.
- k) Temporal refinement enables clustering of spatially overlapping structures, the dissection of initiation point of each cluster in time and following of morphology changes over time all of which are currently infeasible by DBSCAN or other available methods alone.
- l) Morphological fingerprinting for quantification and ranking of interpretable geometric and kinetic descriptors, that to the best of our knowledge is introduced here by us for protein clustering, based on our earlier work on diffusional fingerprinting.

Changes in the manuscript 3.3

The revised manuscript now includes further detail on how SEMORE contributes to the field of machine learning based approaches.

Firstly, how the clustering module of SEMORE extends beyond current state of the art see 2nd paragraph of **Results** :

“The clustering module of SEMORE consists of multiple steps that self-parameterize based on the input data and are scale invariant due to the inherent 3D axis standardization (see Methods). This is designed to account for the inherently heterogeneous nature of biological assembly systems, in size, scale, spatial overlap, density, and morphology as well as the variability across experimental configurations that challenges current analytical tools. The pipeline initially inspects high-density areas in a standardized Euclidean 3D space, using a hyperparameter space pre-defined for this region, and provides an appropriate model choice based on a data-driven decision. The chosen density-based scanning model, either HDBSCAN or DBSCAN, extracts high-density regions of biomolecules (clusters or aggregates) from low-density regions (noise) (Fig. 1b). The initial clustering contains an added topological fail safe to prevent detection of nonsensical structures (see Methods). If a temporal dimension is available, the high-density regions are treated through our temporal refinement (Fig. 1c). Segmentation in time and temporal refinement is strictly required to dissect spatially overlapping structures within high-density areas. The clustered output is further refined by subjecting all identified assemblies to a smart density filter to eliminate falsely predicted assemblies that do not meet an agnostic, data-derived density criteria (Supplementary Fig. 1). The result is a robust clustering model outcompeting current methods and building towards the first general-purpose approach for dynamic SMLM (see Methods and Supplementary fig. 4).”

Secondly, in **Discussion** we elaborate further on the contribution of morphological fingerprinting module to advance ML approaches:

“We envision SEMORE’s application across diverse systems, the continuous extension of the fingerprinting module and the generation of libraries of protein assembly morphologies. Libraries of morphological features could aid mapping of assembly characteristics to their identity and function for advancement of biological understanding, statistical approaches and be the basis for novel supervised learning purposes (Supplementary Fig. 13).”

See also response to comment 3.2 where we highlight the use of supervised learning on the unsupervised output from SEMORE allowing new downstream models to be developed by the community working on the experimental data-derived morphological fingerprints as features

Comment 3.4

Since vibrational and spherical aberrations prove to be a greater hindrance at high resolution. Moreover, live samples are more adversely affected by super-resolution imaging because of high excitation intensity or extended exposure times. Does the SEMORE tool take these concerns into consideration?

Response 3.4

Vibrational and spherical aberrations present large challenges to high resolution imaging especially in live samples, therefore they are important to comment on and we thank the reviewer for bringing this point. Removing such aberrations is often reliant on post-processing. Currently, SEMORE is downstream of any such post-processing steps, as well as the actual detection step as a pipeline working directly on existing localizations to accurately extract underlying structures and provide extensive quantitative features. We acknowledge this was not explicitly discussed in the main text so we have rectified it

Changes in the manuscript 3.4

We now discuss the consequences and the recommended approach of correction before SEMORE or to use SEMORE as an easy approach to extract morphological information quickly before correction. see **Discussion**:

“SEMORE is currently optimized for laterally stable assemblies, albeit the temporal refinement does handle lateral movement below the agnostic, data-derived search range defined per assembly basis (see Methods). Future versions are planned to include our diffusional analysis framework for motion aware clustering analysis. Blinking has insignificant effect on SEMORE analysis, however we recommend common aberrations such as blinking and vibrational or spherical aberrations to be corrected prior to use of SEMORE to avoid any potential misinterpretation of the data.”

Additional minor changes

minor spelling and phrasing mistakes were rectified.

minor addition to account for the fact that we have 5 experimental data sets instead of 2 in the original submission, as well 6 additional ground truth simulated data sets.

addition of a few extra refs to account for reviews comments.

Figures

Response Figure 1: dSTORM data from Nieves et al. ¹ clustered and quantified by SEMORE

Supplementary Figure 17: Evaluation of SEMORE clustering on dSTORM data acquired over several consecutive frames ¹ **a** Raw detections from SMLM data of fibroblast growth receptor 1 (FGFR1) on a MCF7 cell presented by Nieves et al. ¹ **b** Initial clustering by SEMORE's clustering module using the data-driven model choice of HDBSCAN with Min_cluster_size = 15 and Min_sample = 5. Each localization is colored by its SEMORE annotation, with black representing noise and all other colors representing captured clusters. **c** The

final clustering by SEMORE after temporal refinement and smart density filtering (see Methods). Localizations' colour corresponds to the final SEMORE prediction, with black localizations representing predicted clustered suppressed by the smart density filter. Clearly depicting complete extraction of heterogeneous protein clusters while minimizing the inclusion of false positive structures demonstrates the power of temporal refinement, smart filtering and cluster specific re-evaluation. **d** Distribution of detections inside each cluster with each dot representing an identified cluster showing an average of 28 detections and a median of 18 detections. **e** Distribution of area spanned by the detections inside each predicted clusters resulting shows a mean $2232nm^2$ and median of $1829nm^2$. The Relatively small areas compared to the mean of $17000 nm^2$ reported in Nieves et al. ¹ showing the difference in a tight area estimation we define in the morphological fingerprinting (see SI fig. 17) versus a more crude convex hull used in Nieves et al ¹. **f** Density of each cluster for further depth in analysis and heterogeneous nature of protein clusters showing with a mean density of $0.0179 N/nm^2$ and a median of $0.009 N/nm^2$.

Response figure 2: Sx1a-mEos2 sptPALM data from Wallis et al. ³ clustered and temporally quantified by SEMORE.

Supplementary Figure 18: Evaluation of SEMORE on temporarily resolved Sx1a-mEos2 sptPALM data ³.

a, Raw localizations of all detections of Sx1a-mEos2 colored in grey for noise detections and other colors for individual clustered detections captured by SEMORE (HDBSCAN; min_cluster_size: 10, min_samples: 10, cluster_selection_epsilon: 0.04) for the entire field of view (sptPALM traces above length 20) showing near identical results to Wallis et al ³. Red box indicates a region of interest. **b**, Snapshots in time plotting all detections prior to the indicated acquisition time from the region of interest in **a** showing SEMORE captures the time-resolved appearance and growth of clusters. Last snapshot contains a red box highlighting an additional region of interest. **c**, 3D (xyt) plot of the region of interest presented in **b** showing the temporal morphological changes of clustered detections. Red box indicates a hotspot region in space with repeated clustering of detections in time. **d**, Shows the temporal evolution of cluster size for the region of interest in **c** showing SEMORE's successful capture of the repeated clustering of detections also discussed in Wallis et al ³.

Response Figure 4 Evaluation of SEMORE on temporarily resolved live-cell PALM data of ryanodine receptors (RyRs).

Supplementary Figure 19: Evaluation of SEMORE on temporarily resolved live-cell PALM data of ryanodine receptors (RyRs) ⁴. **a**, Raw localizations of all detections of RyRs from live-cell PALM from Hou et al⁴ colored in grey. Scale bar 2 microns. **b**, Initial clustering of data by DBSCAN showing clusters colored by cluster identity. **c**, left panel: Final clustering by the full clustering module of SEMORE utilizing the smart density filtering and temporal refinement with individual clusters colored by identity (DBSCAN; dbsmin_samples: 10, SEMORE: final_min_points: 10, rough_min_points: 10, investigate_min_sample: 10). Hyperparameters of SEMORE chosen to capture the relatively small clusters formed by RyRs. Results show how SEMORE obtains granularity otherwise infeasible by DBSCAN alone. Right panel: Final clusters from SEMORE colored by the “longest shortest distance” feature from the morphological fingerprinting module with clusters exceeding a threshold of 400nm colored in red showing how SEMORE can capture RyR cluster morphology and separate noisy detections. Utilizing the tight packing assumption for RyRs ⁴ identified RyR clusters consist on average of 18 RyR with a median of 14 in line with results from Hou et al ⁴ with slightly larger estimates as we do not filter by intensity.

Response Figure 4: Evaluation of SEMORE' segmentation and structural information extraction on simulated small tetramer assemblies.

Supplementary Fig. 7: Evaluation of SEMORE' segmentation and structural information extraction on simulated small tetramer assemblies.

a) Simulated experiment containing both fibril (orange) and elliptical (blue) morphology class containing 4 points each. Three noise ratios were simulated 1, 2 and 3 (ratio 2 in figure) with 10 simulations per noise levels each with 30 true assemblies. **b)** Assembly extraction by SEMORE's clustering module. SEMORE's hyperparameters were kept across conditions for comparable results and evaluation of automated analysis. **c)** Accuracy evaluation performed on an aggregate-wise level (see methods) grouped into each noise level. SEMORE achieves accuracies of 90% +/- 23% (noise ratio 1), 84% +/- 30% (noise ratio 2) and 77% +/- 37% (noise ratio 3) showcasing the fidelity of the pipeline even at assembly sizes of four detections with three times as many noisy detections. Throughout all 3 noise ratios the median accuracy is 100% indicating that the found assemblies usually contain all the points of the assembly. **d)** The morphological fingerprints extracted for all structures visualized by an out-of-box 2-component UMAP. Right panel shows wrongly classified structures clustered, notably the noise (black) contained in the lower left corner. Each plotted assembly is represented by its noise to signal level (circle: 1, square: 2, triangle: 3) each noise to signal ratio contains 4, 9 and 38 false positives, respectively, compared to >270 true positives. Presence of false positives is due to true assemblies having densities near identical to noise impeding the smart density filter from accurately excluding the noise density. Left panel shows colour coding each assembly by density revealing true assemblies and false positives can be separated by a single feature from morphological fingerprinting. Proving morphological fingerprinting can aid a non-perfect segmentation, allowing for further correction or focus-based re-segmentation.

Response figure 5: SEMORE classification performance for small, sparse clusters of diverse morphologies.

Supplementary Fig. 8: SEMORE classification performance for small, sparse clusters of diverse morphologies.

a) Simulated data containing temporally resolved fibril and static ellipse shape aggregates (coloured orange and blue respectively) containing 8, 15 and 25 points respectively. Each simulated data set contains 30 aggregates equally divided between the two morphology classes, additionally, noise (coloured black) ratios of 1 and 2 (noise ratio 2 depicted in the figure). 10 experiments were simulated for each condition and each noise ratio accumulating to 1800 aggregates over 60 experiments. **b)** Predictions made by SEMORE clustering module, with coloured aggregate being True positives (TP) and black being false positives (FP). SEMORE's clustering module is able to handle the combination of sparse static and temporally resolved assemblies contained within the same experiments. **c)** SEMORE's segmentation performance quantification. Through the SEMORE clustering module an aggregate-wise mean accuracy (see methods) (noise ratio 1 & 2 reported as [1] / [2]) of; 85% / 80% +/- 24% / 31% and a median accuracy of 91%/ 89% for 8 points assemblies, mean 80%/81% +/- 31%/30% with median 94%/94% for 15 point assemblies and a mean 83% / 76% +/- 27%/35% and a median of 95%/93% for 25 point

assemblies. as seen in **d)** the two-component UMAP of morphological fingerprints reveal all false positive assemblies (noise detections) are perfectly separated. Using this separation as correction accuracies improve: mean 87% / 86% +/- 21% / 21% median 91% / 89% for 8 points, 83% / 89 % +/- 27%/20% median 94% / 94% for 15 points and mean 84% / 82% +/- 26% / 27% median 96% / 93% for 25 points. Performance is improved proportional to noise-ratio as the higher noise levels induce more false positives which in turn are corrected by morphological fingerprinting. **d)** Two-component UMAP embedding of the corresponding fingerprints with the different noise-ratio combined within point types (N = 611, 611, 586 for 8, 15, 25 points respectively).

Response Figure 6: Deconvolution of temporal refinement on real and simulated data.

Supplementary Fig. 6: Deconvolution of temporal refinement on real and simulated data.

a) Snapshots of SEMORE's clustering module output at various frames during temporal refinement of found insulin aggregates (also depicted in **Fig. 1C.**) in MinMax transformed space (see Methods). Data displayed in 9 windows equally spaced throughout the 381 frame period, each initially found core point of all dissected aggregates is depicted with a red dot symbolizing the initiation of an aggregate. **b)** The cluster growth is directly visualized in non-transformed space revealing cluster growth in time of **a)** in a 3D plot after the application of the smart-density filtering (see methods). The cone-like shape of the aggregates in the positive time direction demonstrates SEMORE's ability to capture the growth behaviour of assemblies in experimental data. Each depicted cluster has its predicted initiation point marked with projected lines at the bottom of the plot to highlight the vast difference in the onset times of clusters as revealed by SEMORE. **c)** Simulated data (from SI fig. 3) 400 frames after initiation. Data displays that SEMORE can find assembly cores, even with high-size heterogeneity including assemblies as small as 94 points up to 7777 points while also correctly dissecting overlapping assemblies. **d)** The corresponding frame of the predicted initiation was used to estimate the offset in time, from growth start to SEMORE prediction, here plotting ~99% of the distribution. The mean offset of 13 frames, reflects that SEMORE can accurately classify cluster initiation while showcasing the strength of SEMORE in capturing temporal assembly dynamics.

Response Figure 7: Visual representation of SEMORE's ability to capture morphological growth in time

Supplementary Fig. 5: Visual representation of SEMORE's ability to capture morphological growth in time.

(a, b, c) depicts the inherent process of temporal segmentation of assemblies within the clustering module of SEMORE during temporal refinement shown as 9 windows across three morphological classes. **a)** 3 simulated fibrils that spatially overlap during growth are accurately segmented. **b)** 3 simulated isotropic assemblies growing very close to each other accurately extracted and separated by SEMORE. **c)** Simulated growth of 3 sterically-hindered assemblies captured by SEMORE. **a, b, c** showcases the versatility of the general clustering module of SEMORE in capturing temporal evolution of protein assemblies, while general segmentation performance can be seen in main fig 2. The second row (**d, e, f**), shows the structures in 3D with the time being the 3rd axis to visualize assembly growth and morphology evolution. A direct depiction of the predicted assembly initiation point is shown as projected lines to the bottom of the plots revealing SEMORE's ability to keep account of various growth onset times. This is shown for Fibril (**d**), Isotropic (**e**) and sterically hindered growth (**f**) assemblies.

Response Figure 8: Depiction of recurrent SEMORE fingerprinting for dynamic morphology variation of protein clusters.

Supplementary Fig. 15: Depiction of recurrent SEMORE fingerprinting for dynamic morphology variation of protein clusters.

a) Simulated dynamic morphology variation of protein assembly by sequentially exhibiting structures with diverse morphologies. All temporal states consist of 200 points colour-coded by morphology type. Lines projected at the bottom represent a new morphology initiation (coloured by morphology type). For each temporal state, the SEMORE morphological fingerprinting module was used to extract features for probing the morphology changes across temporal states. **b)** The morphological fingerprints were embedded by an out-of-the-box 2-component UMAP resulting in a clear separation of the morphology class across time (with time linked by grey lines). Systems with more or less gradual change of morphology would result in a more or less regressive separation. **c)** MinMax transformed fingerprints plotted into a heatmap with each column representing a temporal state and white lines indicating drastic morphology type changes. Each row represents a morphological feature, in the same order as seen in Supporting Table 1. The heatmap rows showcase the morphological fingerprinting module's ability to capture the mechanistic of temporal states and map the overall features changes involved in major changes in morphology.

Response Figure 9: Benchmarking SEMORE on for heterogeneous, non-uniform noise.

Supplementary Fig 3: Benchmarking SEMORE on for heterogeneous, non-uniform noise.

a) Simulated data used in SI fig. 2, but now exhibiting a heterogeneous noise profile. Heterogeneous noise is generated by introducing 5-25 "noise seeds" each containing 20-50 points have been added to each simulation, additionally, individual Gaussian distributed shifts have been applied to all initial noise points. b) we see SEMORE clustering classification performance as out-of-the-box. c) Evaluating SEMORE clustering performance versus increasing noise ratio (blue area representing biologically relevant noise-levels estimated from real data and with vertical blue line representing insulin experiments). At biologically relevant noise levels or high noise levels accuracies of SEMORE is observed to be above ~90% from noise ratio ~0.2 and onwards with up to 93% at 0.7 noise ratio reflecting SEMORE's ability to separate non-uniform noise also observed in robust treating of experimental insulin data (see main fig. 4). The unrealistic conditions with extremely low noise ratios or no noise at all are associated with lower accuracies as the smart-density noise filter expects noise and removes true positives if no noise is present, which is easily countered by turning it off.

Response Figure 10: Effect of blinking on SEMORE's morphological fingerprinting and characterization of morphological classes.

Supplementary Fig. 9: Effect of blinking on SEMORE's morphological fingerprinting and characterization of morphological classes.

Each row in **a)** and **b)** are grouped and depict a certain simulated structure size (4, 8, 15 and 25 data points, respectively). **a)** Left side shows detections from representative fibril structures as "Ground truth" (orange) prior to blinking and the right side depicts the corresponding structures obtained after blinking (dark red). These post-blinking structures do not include the ground truth points (orange). For each row, 600 ground truth structures were simulated along their corresponding blinking counterpart and as performed in SI fig. 8 two types of morphologies were simulated in equal amounts, namely fibril-like and spherical. **b)** The morphological fingerprints of each structure are visualized by a 2-component UMAP with colour-coded by morphology type. The ground truth structures (no blinking) shown on the left side of **b)** and post-blinking structures on the right side. The results show distortion in separability induced by blinking is positively correlated with the number of detections with a given structure as expected as blinking will dominate the morphology of small structures. From 15 detections in a structure and above blinking has little to no effect on the morphological fingerprinting module of SEMORE.

Response Figure 11: Demonstration SEMORE clustering of morphology shrinkage

Supplementary Fig. 10: Demonstration SEMORE clustering of morphology shrinkage

a) Demonstration of 3 simulated isotropic structures that while proving a challenge due to spatial overlapping also undergo shrinking (colour-coded by identity). **b)** Results show SEMORE accurately identifies spatially overlapping clusters reducing in size at all frames in the simulation allowing the tracking of individual shrinkage in

time. As shrinkage is the direct opposite of growth, SEMORE's clustering module is directly applicable as is after a simple preliminary step of flipping the temporal axis of the simulation or experiment.

Bibliography

1. Nieves, D. J. *et al.* A framework for evaluating the performance of SMLM cluster analysis algorithms. *Nat. Methods* **20**, 259–267 (2023).
2. Lelek, M. *et al.* Single-molecule localization microscopy. *Nat. Rev. Methods Primers* **1**, (2021).
3. Wallis, T. P. *et al.* Super-resolved trajectory-derived nanoclustering analysis using spatiotemporal indexing. *Nat. Commun.* **14**, 3353 (2023).
4. Hou, Y. *et al.* Live-cell photoactivated localization microscopy correlates nanoscale ryanodine receptor configuration to calcium sparks in cardiomyocytes. *Nat. Cardiovasc. Res.* **2**, 251–267 (2023).
5. Zhang, M. *et al.* Direct observation of heterogeneous formation of amyloid spherulites in real-time by super-resolution microscopy. *Commun. Biol.* **5**, 850 (2022).
6. Liu, C. C. *et al.* Robust phenotyping of highly multiplexed tissue imaging data using pixel-level clustering. *Nat. Commun.* **14**, 4618 (2023).
7. Fleming, S. J. *et al.* Unsupervised removal of systematic background noise from droplet-based single-cell experiments using CellBender. *Nat. Methods* (2023) doi:10.1038/s41592-023-01943-7.
8. Raad, J. D. *et al.* Unsupervised abnormality detection in neonatal MRI brain scans using deep learning. *Sci. Rep.* **13**, 11489 (2023).
9. Andersen, C. B. *et al.* Branching in amyloid fibril growth. *Biophys. J.* **96**, 1529–1536 (2009).
10. Xu, C. & Jackson, S. A. Machine learning and complex biological data. *Genome Biol.* **20**, 76 (2019).
11. Thiyagalingam, J., Shankar, M., Fox, G. & Hey, T. Scientific machine learning benchmarks. *Nat. Rev. Phys.* (2022) doi:10.1038/s42254-022-00441-7.
12. Benning, N. A. *et al.* Dimensional Reduction for Single-Molecule Imaging of DNA and Nucleosome Condensation by Polyamines, HP1 α and Ki-67. *J. Phys. Chem. B* **127**, 1922–1931 (2023).
13. Ankerst, M., Breunig, M. M., Kriegel, H.-P. & Sander, J. OPTICS. *SIGMOD Rec.* **28**, 49–60 (1999).
14. A density-based algorithm for discovering clusters in large spatial databases with noise | Proceedings of the Second International Conference on Knowledge Discovery and Data Mining. <https://dl.acm.org/doi/10.5555/3001460.3001507>.
15. Khater, I. M., Nabi, I. R. & Hamarneh, G. A Review of Super-Resolution Single-Molecule Localization Microscopy Cluster Analysis and Quantification Methods. *Patterns (N Y)* **1**, 100038 (2020).
16. Bzdok, D., Altman, N. & Krzywinski, M. Statistics versus machine learning. *Nat. Methods* **15**, 233–234 (2018).

REVIEWER COMMENTS

Reviewer #1 (Remarks to the Author):

I would like to thank the authors for their very thorough and in-depth response to my comments. The extra benchmarking and highlighting the scale invariance has significantly improved the manuscript and will be a very useful tool for SMLM cluster analysis and classification.

I fully recommend the revised manuscript for publication.

Reviewer #2 (Remarks to the Author):

The authors addressed many of my review comments regarding simulation with heterogeneous noise and added the new data analysis results, including time-resolved images. However, unfortunately, some of the newly added data and explanations are not satisfactory enough to resolve my concerns based on the following reasons. Therefore, I strongly request that the authors carefully examine their data and analysis.

1. To address my comments on the presentation of time-resolved image data, the authors added a simulation of dynamic morphology variation of protein assembly in Supplementary Figure 15. However, this simulated data contains only drastic changes between three clearly classifiable different morphologies without transitional states. In contrast, in the real world, we can expect gradual morphological changes in proteins through intermediate states. Therefore, I'm not yet sure whether their method can detect and track such more realistic gradual changes in proteins from the experimental SMLM image data.

2. The authors compared the SEMORE result with previously reported results and asserted that SEMORE provided comparable results with the previous reports. For example, they insisted comparable results with Nieves et al (Supp Fig 17) and Wallis et al (Supp Fig 18). However, I cannot find the related supporting quantitative comparison data, making it difficult to judge whether SEMORE really provides comparable and reasonable results with the reported results. Additionally, the authors insisted that SEMORE could minimize the inclusion of false positive structures compared to HDBSCAN (in Supp Fig 17 experimental data), but I'm uncertain whether the filtered localizations are indeed false positive structures or positive structures that should not be filtered because there is no related data shown. To make a reasonable comparison of their results with others, quantitative comparisons should be provided.

3. In many figures, particularly in supplementary figures, scale bar information or axis units are missing. Consequently, it is challenging for me to judge whether the clustering or filtering is biologically relevant in their image data. Additionally, in supplementary Figure 17 d-f, it appears that the x-axis is meaningless, so a scatter plot for density vs. N or Area would be more reasonable.

4. Although the authors performed additional simulations with various noise variations, I don't understand some of the data. For example, Supplementary Figure 3 shows an increasing accuracy with an increasing ratio of heterogeneous, non-uniform noise, which is contrary to the general expectation. This could be because the data in Supplementary Figure 3c were obtained from just one-time simulation for each noise ratio (there is no error bar in this data), which could not demonstrate the general trend very well. While the authors claimed that the smart-density noise filter is designed to expect noise and remove true positives, such a case could lead to incorrect filtration for real images with low noise as well. Since different SMLM microscopes and different biological samples can produce various noise levels, I believe that if such an effect exists in their analysis method, it should be resolved.

5. Regarding the responses for my fourth comments (2.4), I do understand that unsupervised learning is also a significant component of machine learning. My previous comments might be somewhat unclear in conveying my intended message. During the SEMORE clustering process,

there are several steps involved in clustering meaningful points. The methods you have suggested in this paper to enhance clustering primarily focus on filtering out points and implementing temporal refinements. My point of concern is related to what happens once the filtered and refined points are available. I am uncertain about the contributions of this work to the "machine-learning-based clustering algorithm" at this stage. Even if the filtering and temporal refinements can potentially aid subsequent machine-learning-based clustering algorithms, they are difficult to be considered as a machine-learning algorithm by themselves. Therefore, I am curious if you have made any contributions to the "machine-learning-based clustering algorithm itself" apart from hyperparameter tuning in existing methods like DBSCAN. Examples of such contributions might include modifications to the loss function, alterations to the architecture, improved training techniques, and so on.

Review response 2nd

Review response 2nd	1
Reviewer 1	1
Comment:	1
General response	1
Reviewer 2	2
Comment:	2
General response	2
Comment 2.1:	2
Response 2.1:	2
Changes in manuscript 2.1:	4
Comment 2.2:	4
Response 2.2:	6
Changes in manuscript 2.2:	8
Comment 2.3:	8
Response 2.3:	8
Changes in manuscript 2.3:	9
Comment 2.4:	11
Response 2.4:	11
Changes in manuscript 2.4:	13
Comment 2.5:	13
Response 2.5:	14

Reviewer 1

Comment:

I would like to thank the authors for their very thorough and in-depth response to my comments. The extra benchmarking and highlighting the scale invariance has significantly improved the manuscript and will be a very useful tool for SMLM cluster analysis and classification.

I fully recommend the revised manuscript for publication.

General response

We thank the reviewer for valuable feedback that strengthened the paper and for accepting this for publication

Reviewer 2

Comment:

The authors addressed many of my review comments regarding simulation with heterogeneous noise and added the new data analysis results, including time-resolved images. However, unfortunately, some of the newly added data and explanations are not satisfactory enough to resolve my concerns based on the following reasons. Therefore, I strongly request that the authors carefully examine their data and analysis.

General response

We are sorry to see that newly added biological and simulated data as well as analysis are not satisfactory. We have below fully and in detail addressed the new comments.

Comment 2.1:

1. To address my comments on the presentation of time-resolved image data, the authors added a simulation of dynamic morphology variation of protein assembly in Supplementary Figure 15. However, this simulated data contains only drastic changes between three clearly classifiable different morphologies without transitional states. In contrast, in the real world, we can expect gradual morphological changes in proteins through intermediate states. Therefore, I'm not yet sure whether their method can detect and track such more realistic gradual changes in proteins from the experimental SMLM image data.

Response 2.1:

We value the reviewer's feedback towards creating the best and most convincing display of SEMORE's capabilities. We agree that Supplementary figure 15 currently shows a protein assembly undergoing more drastic changes. We displayed this, along with other supplementary figures, as the simplest case to show SEMORE captures changes in morphology over time.

To execute directly on this revised feedback, we evaluated below SEMORE's capacity to characterize 8900 intermediate structures of gradual dynamic morphology variation in supplementary figure 26. The figure displays 90 structures from the three distinct morphologies sequentially, yet no two similar classes in a row, along with 100 interpolated structures as intermediates forming the transition between each of the 90, thus totalling 8900 structures. SEMORE accurately captures intermediate morphology classes and tracks their gradual dynamic morphology putting intermediate structures on a gradient between the distinct groups of where the transition is happening. Given these structures exist in a continuous space SEMORE allows users to decide on decision boundaries for each of the structures.

We wish to stress that a central element of SEMORE is that it inherently performs temporal segmentation, thus capturing gradual changes. As detailed in the original submission SEMORE will segment and quantify each frame individually in a given data set, thus each identified cluster will have a temporal segmentation capturing its changes in time. Please also notice that aggregation growth is an inherently gradual change in morphology. We had displayed in main figure 1, 2, and 3, as well as, Supplementary figure 4, 5, and 6) of the original submission SEMORE's capacity to capture growth.

Supplementary Fig. x: SEMORE's Morphological Fingerprint captures gradual transitions in morphology.

Thirty aggregates of each morphology class (fibril-like: fib, isotropic: iso, sterically-hindered/random: rand) of equal size were simulated resulting in ninety distinct structures which are placed sequentially in random order whilst ensuring no consecutive types. Between each of the ninety structures (89 transitions) 100 positions are constructed from interpolation (see methods). Thus, a total of 8900 intermediates gradually changing morphology between the three structures were evaluated using the fingerprinting module of SEMORE. All resulting transitions are connected

in time, meaning the final structure of the (i) transition is the starting structure of (i+1) transition, thus creating a single dynamic structure of gradual transitions. **a-c** Shows representative transitions from fibril to isotropic, from fibril to random, and from random to isotropic, respectively. The lines drawn between points in **a-c** represent the 100 different positions expressed throughout the interpolation from the initial structure to the target structure. The color gradient of the box around each panel in **a-c** is the RGB representation of the percentage of morphology transition between the three distinct morphology classes. This color map is also used in **d**. (**d**) To visualize the general data manifold of the morphological fingerprints across the 8900 structures, the high dimensional morphological feature set was dimensionality reduced through a UMAP (n_components : 2, n_neighbors: 400, min_distance: 0.5). Similar to supplementary fig. 15 the embedding shows capture of distinct morphology classes and now also the continuous gradual change between the distinct morphology classes, as each structure morphs into the next. Thus, highlighting SEMORE's strength in capturing gradual morphological evolution. 97.9% of data are grouped in the continuous flow between the three distinct morphologies ($x=7.5, y=6$). The small group around the ($x=-5, y=0$) area contains 186 points representing 2.1% of the data, although outliers still show a continuous flow between morphology classes. From this visual inspection, it is clear that the fingerprints capture both drastic and gradual changes in structure.

Changes in manuscript 2.1:

To fully address the comment of the referee we have:

a) added a new supplementary figure 26 detailing the dynamic morphology variation and how SEMORE track this

b) added a paragraph in the section "**Morphological fingerprinting captures defining features separating heterogeneous assemblies.**"

"A central element of SEMORE is that it inherently performs temporal segmentation thus offering the potential to capture gradual morphological changes in super resolution data. To evaluate SEMORE's performance we simulated dynamic morphology variation between three major morphologies (fibril-like, isotropic, and sterically-hindered) (Supplementary Figs. 15 and 26). We simulated thirty structures of each morphology class (totalling 90), placed these sequentially in random order whilst ensuring no identical morphology consecutively. Between each of the ninety structures 100 positions are constructed by interpolation (see Methods) resulting in 8900 intermediate structures. UMAP representation in Supplementary Fig 26 shows SEMORE accurately captures distinct morphology classes and reliably tracks their gradual dynamic morphology change by placing intermediate structures on a gradient between the distinct morphology classes of where the transition is happening. Note, the UMAP is simply a visualization tool to show structure of the high dimensional data manifold of the 8900 morphological fingerprints, it is not a requirement for usage and may vary for specific cases. As these structures exist in a continuous space SEMORE allows future users to identify dynamic morphological variations and decide on system-specific decision boundaries for each of the structures. "

Comment 2.2:

2. The authors compared the SEMORE result with previously reported results and asserted that SEMORE provided comparable results with the previous reports. For example, they insisted comparable results with Nieves et al (Supp Fig 17) and Wallis et al (Supp Fig 18). However, I cannot find the related supporting quantitative comparison data, making it difficult to judge whether SEMORE really provides comparable and reasonable results with the reported results. Additionally, the authors insisted that SEMORE could minimize the inclusion of false positive structures compared to HDBSCAN (in Supp Fig 17 experimental data), but I'm uncertain whether the filtered localizations are indeed false positive structures or positive structures that should not be filtered because there is no related data shown. To make a reasonable comparison of their results with others, quantitative comparisons should be provided.

Response 2.2:

We thank the reviewer for allowing us to clarify and improve the presentation of the additional experimental data we use to showcase the operational utility of SEMORE. The quantification of

SEMORE performance is extensively validated on simulated data where there is undoubtedly ground truth (supplementary figs 4, 11, 13, etc.). Comparison of performance on experimental data however requires ground truth which is only attained by field specific expert knowledge. While we had already some quantification, as we detail below, we felt that directly and outloud comparing SEMORE with existing methods dealing with experimental data where there is no strictly defined ground truth might be misinterpreted as criticism to our esteemed colleagues and future collaborators. Therefore, we had kept it minimal in the original manuscript especially as methods and research goals are different. To fully address the comment of the reviewer we now provide below the requested additional quantitative comparison.

SI figure 17.

- Supplementary figure 17 included reported area per cluster as Nieves et al and we had even chosen the same plotting style for easier comparison. We had also emphasized that the two approaches calculate area differently, i.e., convex hull counting white space (Nieves et al) vs Delaney triangulation and edge pruning to avoid redundant white space (ours). The comparison of the two methods was already displayed in Supplementary figure 20.

To further strengthen the quantitative comparison, we provide:

- a) a comparison of Nieves et al and SEMORE that show **an 88% agreement in categorizing detections as noise**, showing that SEMORE and Nieves et al have agreement in which detections are noise and which are signal.
- b) a raw count of the number of protein assemblies identified by SEMORE across the 15 experiments provided in Nieves et al and compared to the declared results in Nieves et al. We report that **SEMORE identifies 82% of clusters found by Nieves et al** (130 out of 158 assemblies) thus supporting SEMORE obtains comparable results, while also providing morphological fingerprints of each assembly.

Importantly, Nieves et al use a DBSCAN with parameters ($\text{eps} = 65$, $\text{minPts} = 32$) chosen based on highest spatial similarity to one of 10 simulated data scenarios, such settings disfavor smaller clusters thus have inherent tendency to make larger clusters. SEMORE self-parameterized based on the experimental data itself and it includes a round of refinement allowing the capture of smaller clusters and splitting of large into distinct smaller clusters (as seen in main figure 1 and the updated supplementary figure 17). SEMORE captures 9.4 new small clusters per movie that is classified as noise by Nieves et al. SEMORE can easily report the same clusters by turning off the additional refinement and only using the DBSCAN. We stress that experimental data have no ground truth and they are reliant on field expertise, so we would refrain from elaborating in the main text on why SEMORE is better and rather state that the results are similar. Lastly we point out that the comparison of DBSCAN and SEMORE on simulated, and thus ground truth data, is shown in supplementary figure 4 where SEMORE also exhibited clear superiority in detecting clusters and excluding false positives.

Supplementary Fig. 17 (UPDATED): dSTORM data from Nieves et al.¹ clustered and quantified by SEMORE
a Raw detections from SMLM data of fibroblast growth receptor 1 (FGFR1) on a MCF7 cell presented by Nieves et al.¹
b Initial clustering by SEMORE's clustering module using the inherent data-driven model choice of HDBSCAN (Min_cluster_size = 15 and Min_sample = 5). Each localization is colored by its SEMORE annotation, with black representing noise and all other colors representing captured clusters.
c The final clustering by SEMORE after refinement and smart density filtering (see Methods). Localizations are colored corresponding to the final SEMORE prediction. Red box depicts the same zoom-in as seen in Nieves et al inside which they report 2 unique clustering indices similar to the initial clustering seen in **b**. SEMORE's additional refinement split one of these into four distinct clustered indexes, thus resulting in five clusters. Keeping these settings constant through the 15 datasets from Nieves et al.
d Histogram of cluster area estimation. SEMORE identifies mean cluster area of $0.004 \pm 0.003 \mu m^2$ and median of $0.003 \mu m^2$. SEMORE provides more fine-grained area calculation than convex hull (see supplementary figure 20) and may split larger clusters to smaller during rounds of refinement.
e Distribution of points contained in the proposed clusters with a mean of 52 ± 108 and a median of 29. SEMORE and Nieves et al. obtain qualitative similar results and have 88% agreement in assigning points as noise showing the general agreement as methods

To answer the second comment of the reviewer on minimizing false positives in supplementary figure 17. We assume the reviewer means to compare to Nieves although they use DBSCAN ("the authors insisted that SEMORE could minimize the inclusion of false positive structures compared to HDBSCAN (in Supp Fig 17 experimental data)").

- As attend above, Nieves et al and SEMORE have 88% agreement in categorizing detections as noise
- Nieves et al color true and false positives, thus visual inspection of SEMORE's prediction and the declared true and false positives from Nieves et al provide a very clear qualitative estimate of the correspondence. We stress again that since these are experimental data it is hard to strictly define ground truth. We have updated the figure to show the exact

same field of view as Nieves et al (figure 5 in Nieves et al) to facilitate an easier visual comparison and provided 88% agreement in categorizing detections as noise showing that the reported results are concordant.

SI figure 18.

Supplementary figure 18 included a direct comparison of the specific example from Wallis et al showing a cluster being captured three times across time and SEMORE capturing the exact same as seen in SI figure 18c-d.

To provide a further quantification we counted the number of clusters reported in a crop-out from Wallis et al and compared to the raw cluster count identified by SEMORE. **Wallis et al have 104 clusters while SEMORE has 107** (See the figure below). We stress that SEMORE not only segments clusters but inherently tracks and links clusters in time for tracking morphology over time. Therefore, we manually counted the areas of interest identified by Wallis et al and by SEMORE in a single frame.

SI figure 19

Lastly, we are pleased that the results in SI figure 19 were satisfactory and without comments as this example also highlights the ability to generalize across experimental conditions as visually and quantitatively validated to results of Hou et al. To be consistent we also provide a direct comparison demonstrating once more SEMORE is on par with existing toolboxes.

Changes in manuscript 2.2:

To fully address the comment of the reviewer we

- a) Updated and modified figure text of SI figure 17.

- b) Added the number of identified clusters to supplementary figure 18. “Last snapshot shows the N=107 for the total identified clusters in this snapshot and contains a red box highlighting an additional region of interest.”
- c) Added in supplementary figure 19 an extra sentences for the quantitative comparison “ Utilizing the tight packing assumption for RyRs ³ identified RyR clusters SEMORE reports on average of 18 RyR with a median of 14 RyR in line with the 9 RyR reported by Hou et al ³ importantly achieving so without any intensity thresholding.”
- d) Added in section “**Precise extraction and quantification of experimental super-resolution data.**” “While in experimental data sets it is hard to strictly define ground truth SEMORE outputs qualitatively outputs identical predictions with current states of the art.”

Comment 2.3:

3. In many figures, particularly in supplementary figures, scale bar information or axis units are missing. Consequently, it is challenging for me to judge whether the clustering or filtering is biologically relevant in their image data. Additionally, in supplementary Figure 17 d-f, it appears that the x-axis is meaningless, so a scatter plot for density vs. N or Area would be more reasonable.

Response 2.3:

We agree with the reviewer’s suggestion towards remaking supplementary Figure 17 d-f to be more easily readable. We were following the plotting template of Nieves et al for a more direct and easier comparison (see answer to reviewer comment 2.2) but following the reviewer's suggestion that the x-axis is meaningless we move away from Nieves et al plotting approach and now display histograms in the revised supplementary fig. 17.

The reviewer also commented: “*scale bar information or axis units are missing. Consequently, it is challenging for me to judge whether the clustering or filtering is biologically relevant in their image data.*”

We fully agree that the scale bar is instrumental for judging dimensions of the morphologies.

In the initial response to reviewer 1 we had already explained that SEMORE is purposefully designed to be dimension-independent, therefore the simulated data are without a scale bar, to which they responded: “*The extra benchmarking and highlighting the scale invariance has significantly improved the manuscript and will be a very useful tool for SMLM cluster analysis and classification. I fully recommend the revised manuscript for publication.*”

Please see the full question and response at the end of this section in blue colour fonts.

However, we realize that snapshots from the data in main figure 4 seen in supplementary figures 24 and 25 did not contain error bars and only the full scale figure 4 did. We have now rectified this and we thank the reviewer for noticing this.

As we detail below the scale invariance of SEMORE is due to our choice of 3D axis standardization. Working with standardized dimensions enables SEMORE to work across experimental configurations and dimensionalities. This allows SEMORE to operate independently of the imaging dimension and achieve high classification accuracies solely based on the number of localizations contained within the protein assembly given the spanning area.

Therefore, the simulated data is of arbitrary units, and we did not include a scale bar, however we have now included scale bars with “a.u” for units based on the reviewer’s comment.

We emphasize that SEMORE proves its dimension independence by accurate segmentation across the 5 diverse biological systems spanning from nanometer to micrometer and further by extensive quantification on simulated data at different scales. This should also help the reviewer to “judge whether the clustering or filtering is biologically relevant in their image data”. The simulated data included in the manuscript supports the experimental data and serves as ground truth quantifying the output of SEMORE.

Changes in manuscript 2.3:

- 1) Acknowledging the scale invariance may not be clear to the audience we have added an extra short section in results section that already discusses the scale invariance :
“This allows SEMORE to operate independently of the imaging and cluster dimensions and achieve high classification accuracies solely based on the number of localizations contained within the protein assembly given the spanning area. “
- 2) We have added the histograms in Supplementary figure 17
- 3) We have added scale bars on a.u in the simulated data in Supplementary figures 1, 2, 13, 20, 21 and scale bars at the experimental snipsets displayed in Supplementary figures 24, 25.
- 4) We further refer to the comment and response of the first round of review (in blue below):
- 5) We add in the discussion the key result that SEMORE agnostically segments across five experimental data sets spanning nanometers to micrometers by modifying an existing sentence .” Effectively, SEMORE operates across five experimental data sets with spatial dimensions spanning 3 orders of magnitude from nanometer⁵⁸ to micrometers¹⁹ and temporal dimensions spanning from milliseconds⁶⁰ to second¹⁹. This fact provides strong support for SEMORE as a universal, input-independent model for the SMLM community to use in conjunction with, or to be incorporated into SMAP, or as a convenient standalone toolbox”

Comment 1.1 (reviewer 1, first round)

The clusters within the simulated data seem to be very large in scale, some on the order of 10 microns. This is likely to be an uncommon scenario in biological SMLM data, especially when it comes to clustering of that data. I would recommend some simulations on the scale of the NPC data, i.e., several clusters sub-micron with diverse morphology. A recent paper cited in the manuscript proposed several different clustering scenarios at this scale (Nieves et al., NatMeth, 20, pages 259–267, 2023) where two different cluster types are present within the region. This would be really nice to see clusters of similar density (in that paper it was approximately 10-20 per 3x3 micron²), but analysed in a similar fashion here, i.e., analysis of the whole 40x40 micron region. This would be quite powerful if whole fields of view could be analysed quickly, without the need for subdivision of the data (see later point on performance).

Response 1.1

The reviewer correctly points out that large aggregates are generally depicted in the manuscripts. We stress here that SEMORE clustering module is scale-invariant and solely relies on localisations. This is due to our choice of 3D axis standardization. Working with standardized dimensions enables SEMORE to work across experimental configurations and dimensionalities. This allows SEMORE to operate independently of the imaging dimension and achieve high classification accuracies solely based on the number of localizations contained within the protein assembly given the spanning area. The comment still raises a very valuable point on the generalization and performance of SEMORE and to demonstrate this and fully answer the comment of the reviewer we performed two studies:

- 1) Simulated diverse, nanometer dimension structures, and quantified SEMORE performance on their segmentation and classification for data with 8, 15 and 25 localisations in an area that corresponds to ~1µm. As can be seen in Supplementary Fig. 8 (Response figure 5) in the revised version, SEMORE reached an accuracy of >90% in extracting clusters with 8 or more localisations with the morphology fingerprinting completely separating true clusters from noise.
- 2) Simulated protein assemblies of just 4 detections and evaluated SEMORE's performance on such a small cluster representing the minimal representation of a tetramer. SEMORE achieves accuracies of up to 70-90% in segmenting these tiny structures depending on the noise levels while the morphological fingerprinting module captures these noisy detections allowing for potential post-processing and accuracy increase, Supplementary Fig. 7 (Response figure 4).
- 3) Evaluated the number of points needed for SEMORE to accurately detect the initiation points of spatially overlapping clusters. The high accuracy of SEMORE to capture initiation points can be quantified and visualized using simulated data showing is just ~13 frames off on average across 3 types of spatially overlapping morphologies. Furthermore, qualitative assessment using experimental data of temporally resolved insulin aggregation supports the reliability of SEMORE, Supplementary Fig. 5+6, response figure 6+7).

In essence, ~10 detections is enough for SEMORE to obtain high segmentation and classification accuracy which is well below the common number of detections in most SMLM experiments. We analyze dSTORM data of Nieves et al. NatMeth, 20, pages 259–267, 2023¹ obtaining similar densities and cluster sizes but in fact analyzing the entire field of view at once see new SI fig 17 (response figure 1). We further use the approach from Nieves et al., NatMeth, 20, pages 259–267 to simulate structures with blinking, see comment 1.3.

Changes in the manuscript 1.1

To fully address the comment of the reviewer in the revised version of the manuscript we have added a new section within **“Accurate extraction of individual assemblies across diverse biologically inspired growth types.”**:

“To further evaluate the performance of SEMORE on segmentation and analysis of dynamic SMLM data we performed a series of stress tests on simulated data. We firstly evaluated SEMORE's ability to track morphological changes in time using simulated data of 3 types of spatially overlapping protein clustering morphologies with temporal information included. Snapshots of SEMORE's clustering in time provide visual confirmation of SEMORE's ability to track morphological changes in time (see Supplementary Fig. 5 for snapshots of simulated and Supplementary Fig. 6 for experimental data). Further quantification of SEMORE's ability to accurately track and segment spatially overlapping protein assemblies in time reveals growth onset times across 3 morphological classes are predicted with an average offset of just ~13 frames (Supplementary Fig. 6). Subsequently, we test the clustering module on

simulated, sparse structures containing as little as 4, 8, 15 and 25 point detections akin to data of protein oligomerization (see Supplementary Fig. 7 & 8). SEMORE analyzes the entire field of view at once and extracts structures down to 4 detections while maintaining >90% accuracy at biologically relevant noise to signal ratios. The morphological fingerprinting module can further refine the false positive detections from noise with as little as 4 detections, achieve full separation from noise at 8 detections and classify morphological classes in true structures at 15 detections (see Supplementary Fig. 8 & 9). Lastly we evaluated its performance to degenerative structures that shrink in time, akin to protein de-polymerisation. SEMORE accurately segments 3 anisotropically degenerative morphologies showcasing it can be used to analyze dynamic shrinkage or depolymerisation of protein clusters (see Supplementary Fig. 10). In essence SEMORE only requires <10 detections to accurately segment and classify heterogeneous structures with dynamic morphologies further demonstrating its operational utility and potential for 4D SMLM.”

Elaborated in **results** on the scale invariance:

“The clustering module of SEMORE consists of multiple steps that self-parameterize based on the input data and are scale invariant due to the inherent 3D axis standardization (see Methods). This is designed to account for the inherently heterogeneous nature of biological assembly systems, in size, scale, spatial overlap, density, and morphology as well as the variability across experimental configurations that challenges current analytical tools.”

In addition, to address the comment we have added 5 new SI figures, namely Response figures: 1,4,5, 6, and 7.

Comment 2.4:

4. Although the authors performed additional simulations with various noise variations, I don't understand some of the data. For example, Supplementary Figure 3 shows an increasing accuracy with an increasing ratio of heterogeneous, non-uniform noise, which is contrary to the general expectation. This could be because the data in Supplementary Figure 3c were obtained from just one-time simulation for each noise ratio (there is no error bar in this data), which could not demonstrate the general trend very well. While the authors claimed that the smart-density noise filter is designed to expect noise and remove true positives, such a case could lead to incorrect filtration for real images with low noise as well. Since different SMLM microscopes and different biological samples can produce various noise levels, I believe that if such an effect exists in their analysis method, it should be resolved.

Response 2.4:

We appreciate the feedback for the missing clarity in Supplementary Fig. 3 and for allowing us to rectify this. The current supplementary fig. 3 relies on the data displayed in Supplementary fig 2, which, as stated in the figure legend, is a summary across 5 separate simulations each containing 13 protein assemblies. In addition, we do display in both Supplementary figures 2 and 3 an error-metric across the 5 simulations by showing the variance of accuracy, i.e. the squared standard deviation (orange line at ~12%). We thank the reviewer for pointing out that this was not clear enough and we have now further clarified this in the figure legend.

The reviewer also commented on an upward trend in accuracy. We agree that the first 2 of 16 points display an upwards trend until the accuracy curve flat lines from point 3 to 16. These 2 points are associated with both the largest variation in accuracy and the lowest accuracy. In addition, these points represent experimental conditions with extremely low noise or no noise at all, unlikely for any single molecule localisation experiment. We had commented already on this in the figure legend stating that “at unrealistically low noise levels the smart density filter expects noise and removes true positives if no noise is present, and that this can be easily turned off.”

We emphasize that the smart noise filter only lowers accuracy for the case of no noise or noise being 10 times less than signal. The filter is easily turned off for users operating with systems that do not produce any noise or next to no noise, albeit we are unaware of such technological development. A realistic noise level is displayed by the blue shaded area. While we were aware of the above, we kept the filter on for transparency into SEMORE's output.

To address the reviewer comment and alleviate any confusion we ran SEMORE without the noise filter (see new panel in Supplementary fig 3). As expected, not utilizing smart density filtering results at better accuracy at unrealistically low noise levels, but accuracy declines at, and above, realistic noise levels. Importantly and directly addressing the reviewer's concern SEMORE's segmentation accuracy at the extremely low or no noise levels joined the high accuracy displayed at the more realistic noise cases.

Supplementary Fig 3: Benchmarking SEMORE on for heterogeneous, non-uniform noise.

a) Simulated data as described and used in SI fig. 2, that includes 5 separate simulations with 13 protein assemblies each, but now exhibiting a heterogeneous noise profile. Heterogeneous noise is generated by introducing 5-25 "noise seeds" each containing 20-50 points have been added to each simulation, additionally, individual Gaussian distributed shifts have been applied to all initial noise points. b) we see SEMORE clustering classification performance as out-of-the-box. c) Evaluating SEMORE clustering performance versus increasing noise ratio (blue area representing biologically relevant noise-levels estimated from real data and with vertical blue line representing insulin experiments). At biologically relevant noise levels or high noise levels accuracies of SEMORE is observed to be above ~90% with variance accuracy of ~12% from noise ratio ~0.2 and onwards with up to 93% at 0.7 noise ratio reflecting SEMORE's ability to separate non-uniform noise also observed in robust treating of experimental insulin data (see main fig. 4). The unrealistic conditions with extremely low noise ratios or no noise at all are associated with lower accuracies as the smart-density noise filter expects noise and removes true positives if no noise is present, which is easily countered by turning it off. d) SEMORE's clustering performance on the same data set without the smart noise filter (blue area representing biologically relevant noise-levels estimated from real data and with vertical blue line representing insulin experiments). SEMORE accuracy is above ~90% from conditions with no noise up to around noise ratios of ~1.

Turning off the noise filter results in better performance at extremely and unrealistically low or no noise ratios (zero noise to noise-to-signal ratio of 0.1), but the performance strongly declines starting from around biologically relevant noise levels. For users operating with little or no noise in their experimental system the results indicate the smart filter should be turned off while users operating with realistic noise are recommended to utilize the full SEMORE pipeline.

Changes in manuscript 2.4:

1. We highlight in the text that users should evaluate performance on their given system without the noise filter and turn smart density filtering off, only at unrealistically low noise levels, adding in the 2nd paragraph of section “**Accurate extraction of individual assemblies across diverse biologically inspired growth types.**”:

While at unrealistically low noise ratios i.e ten times lower than signal, smart density filter can result in removal of true positives, we recommend using the full SEMORE pipeline for data with noise (Supplementary Fig. 3)” .

2. Added a new panel in supplementary fig 3 showing SEMORE’s performance without the noise filter.

3. We emphasize in the figure text of SI figure 3 that it is based on 5 simulated data sets with 13 protein assemblies each.

Comment 2.5:

5. Regarding the responses for my fourth comment (2.4), I do understand that unsupervised learning is also a significant component of machine learning. My previous comments might be somewhat unclear in conveying my intended message. During the SEMORE clustering process, there are several steps involved in clustering meaningful points. The methods you have suggested in this paper to enhance clustering primarily focus on filtering out points and implementing temporal refinements. My point of concern is related to what happens once the filtered and refined points are available. I am uncertain about the contributions of this work to the "machine-learning-based clustering algorithm" at this stage. Even if the filtering and temporal refinements can potentially aid subsequent machine-learning-based clustering algorithms, they are difficult to be considered as a machine-learning algorithm by themselves. Therefore, I am curious if you have made any contributions to the "machine-learning-based clustering algorithm itself" apart from hyperparameter tuning in existing methods like DBSCAN. Examples of such contributions might include modifications to the loss function, alterations to the architecture, improved training techniques, and so on.

Response 2.5:

Thank you for allowing us to elaborate further on this. We would like to emphasize two key points: Firstly, the reviewer states that “What happens once the filtered and refined points are available” It is important to note that without the developed modalities of SEMORE, (Automatic, data-driven model selection, semi-supervised hyperparameter choices based on experimental data, topological failsafe, Smart noise filtering and temporal refinement,) the correct clustering and segmentation of experimental data and tracking its morphology in time is not easily available, see main figure 1b vs 1d or SI figure 4. Effectively the SEMORE pipeline, building on existing models, contributes to the field by providing a universal automated and self-parameterized pipeline efficiently resolving a major obstacle that challenges the widespread implementation of SMLM, that of analysis and segmentation of the data.

The reviewer also comments “*Even if the filtering and temporal refinements can potentially aid subsequent machine-learning-based clustering algorithms...they are difficult to be considered as a machine-learning algorithm by themselves.*” We stress here that the refinement and filtering do not all happen prior to DBSCAN or HDBSCAN but rather as rounds of refinement in the pipeline. These rounds are both before and after the initial clustering by current methods, although mostly after. We are thus confused as to how such a scenario should be imagined, but if users somehow already have their refined and filtered points available, we contribute with the morphological fingerprinting module for further quantification.

The reviewer continues : “*... Therefore, I am curious if you have made any contributions to the "machine-learning-based clustering algorithm itself" apart from hyperparameter tuning in existing methods like DBSCAN...*”. We wish to highlight a few papers in interdisciplinary journals²⁻⁵ that present methods built on existing foundational tools repurposed to work for specific fields. Whether contributions are contributions to the "machine-learning-based clustering algorithm itself" appears more semantic of nature rather than objective. For instance, Chen et al ⁴ and Walker et al ³ both enriched an existing machine learning tool, also DBSCAN, with approaches such as filtering, Otsu’s algorithm, constructing graphs etc. to obtain results otherwise not readily unobtainable. Likewise, we enrich an existing tool that could not function properly for super-resolution data, extract the results of the manuscript, and do so across diverse experimental conditions and biological systems. Importantly, our method automates the agnostic analysis of super-resolution data across experimental conditions and biological systems as clearly demonstrated by the application to 5 diverse experimental datasets.

We could further argue that the original DBSCAN/(HDBSCAN) optimizes some objective function $f(x)$ but we optimize $g(f(x))$ where g is our contributions working both prior and after the output of DBSCAN. In fact, unsupervised machine learning clustering in general groups points based on characteristics by optimizing some objective function. Our contribution effectively changes these characteristics, the pipeline and thus the outcome of the ML model. This can be seen as either changing the training approach or changing the architecture/pipeline.

We are in fact carefully phrase our contribution as a new approach to analyze super-resolution data that enables analysis across experiments and systems, something that was easily accessible before (see also main figure 2 and Supplementary figures 4, and 19 for DBSCAN and SEMORE on simulated experimental data). “ SEMORE, a semi-automatic machine learning framework for universal, system and input-dependent, analysis of super-resolution data”

Bibliography

1. Nieves, D. J. *et al.* A framework for evaluating the performance of SMLM cluster analysis algorithms. *Nat. Methods* **20**, 259–267 (2023).
2. Yu, W., He, B. & Tan, K. Identifying topologically associating domains and subdomains by Gaussian Mixture model And Proportion test. *Nat. Commun.* **8**, 535 (2017).
3. Walker, B. L. & Nie, Q. NeST: nested hierarchical structure identification in spatial transcriptomic data. *Nat. Commun.* **14**, 6554 (2023).

4. Chan, H. *et al.* Machine learning coarse grained models for water. *Nat. Commun.* **10**, 379 (2019).
5. Mulhall, E. M. *et al.* Direct observation of the conformational states of PIEZO1. *Nature* **620**, 1117–1125 (2023).

REVIEWERS' COMMENTS

Reviewer #2 (Remarks to the Author):

The authors addressed all the concerns I had regarding the previous version of the manuscript. I appreciate that the authors went the extra mile to address the various concerns.